RESEARCH COMMUNICATION

# Dopamine neuron glutamate cotransmission evokes a delayed excitation in lateral dorsal striatal cholinergic interneurons

Nao Chuhma[1,2]*, Susana Mingote[1,2], Leora Yetnikoff[1,3,4], Abigail Kalmbach[1,2,5], Thong Ma[6], Samira Ztaou[1,2], Anna-Claire Sienna[1,2], Sophia Tepler[1,2], Jean-Francois Poulin[7], Mark Ansorge[2,5], Rajeshwar Awatramani[7], Un Jung Kang[6], Stephen Rayport[1,2]*

[1]Department of Molecular Therapeutics, New York State Psychiatric Institute, New York, United States; [2]Department of Psychiatry, Columbia University, New York, United States; [3]Department of Psychology, College of Staten Island, New York, United States; [4]CUNY Neuroscience Collaborative, The Graduate Center, City University of New York, New York, United States; [5]Department of Developmental Neuroscience, New York State Psychiatric Institute, New York, United States; [6]Department of Neurology, Columbia University, New York, United States; [7]Department of Neurology, Northwestern University, Chicago, United States

*For correspondence:
nc2027@columbia.edu (NC);
sgr1@columbia.edu (SR)

Competing interests: The authors declare that no competing interests exist.

**Abstract** Dopamine neurons have different synaptic actions in the ventral and dorsal striatum (dStr), but whether this heterogeneity extends to dStr subregions has not been addressed. We have found that optogenetic activation of dStr dopamine neuron terminals in mouse brain slices pauses the firing of cholinergic interneurons in both the medial and lateral subregions, while in the lateral subregion the pause is shorter due to a subsequent excitation. This excitation is mediated mainly by metabotropic glutamate receptor 1 (mGluR1) and partially by dopamine D1-like receptors coupled to transient receptor potential channel 3 and 7. DA neurons do not signal to spiny projection neurons in the medial dStr, while they elicit ionotropic glutamate responses in the lateral dStr. The DA neurons mediating these excitatory signals are in the substantia nigra (SN). Thus, SN dopamine neurons engage different receptors in different postsynaptic neurons in different dStr subregions to convey strikingly different signals.
**Editorial note:** This article has been through an editorial process in which the authors decide how to respond to the issues raised during peer review. The Reviewing Editor's assessment is that all the issues have been addressed (see decision letter).
DOI: https://doi.org/10.7554/eLife.39786.001

## Introduction

Dopamine (DA) neurons send dense projections to the striatum (Str) that are topographically organized. Medially located ventral tegmental area (VTA) DA neurons project to the ventromedial Str; more laterally located substantia nigra (SN) DA neurons project to the dorsolateral Str (*Haber et al., 2000*; *Ikemoto, 2007*). DA neuron gene expression profiles identify DA neuron subtypes differentially distributed within the ventral midbrain (*Poulin et al., 2014*) that likely project differentially to striatal subregions, as well as extrastriatal regions. DA neurons are heterogeneous in their

membrane excitability and modulation (*Morales and Margolis, 2017*; *Roeper, 2013*), and the synaptic signals they convey (*Chuhma et al., 2017*).

There is significant heterogeneity in DA release and its modulation in different Str subregions (*Sulzer et al., 2016*). DA neurons make fast D2-mediated DA synaptic connections to cholinergic interneurons (ChIs) in the dorsal Str (dStr) (*Chuhma et al., 2017*), while they elicit slower DA actions via volume transmission throughout the Str (*Surmeier et al., 2014*; *Tritsch and Sabatini, 2012*). DA neuron glutamate cotransmission contributes striking regional heterogeneity; in the nucleus accumbens (NAc) medial shell, DA neurons elicit glutamate EPSCs in the three major Str cell types, with the strongest in ChIs (*Chuhma et al., 2014*). In the dStr, DA neuron glutamate EPSCs have not been seen in ChIs (*Chuhma et al., 2014*), while they have been variably observed in dStr spiny projection neurons (SPNs) (*Mingote et al., 2015*; *Stuber et al., 2010*; *Tritsch et al., 2012*).

The dStr comprises multiple functional subregions, determined by cortical inputs (*Hunnicutt et al., 2016*; *Joel and Weiner, 2000*; *Liljeholm and O'Doherty, 2012*). The medial and lateral subregions correspond roughly to associative and sensorimotor Str, respectively (*Joel and Weiner, 2000*; *Kreitzer, 2009*; *Liljeholm and O'Doherty, 2012*; *McGeorge and Faull, 1989*; *Voorn et al., 2004*). DA modulates learning and performance of goal-directed behavior in the medial dStr (mdStr), and formation of habits in the lateral dStr (ldStr) (*Faure et al., 2005*; *Hilário and Costa, 2008*; *Lerner et al., 2015*). Studies of functional synaptic connectivity have addressed differences between the NAc and dStr, but not between dStr subregions. Whether heterogeneity in DA neuron synaptic actions extends to dStr subregions has not been elucidated.

We have compared synaptic responses elicited by DA neurons in identified mdStr and ldStr neurons, focusing principally on synaptic connection to ChIs, since the most prominent direct DA neuron synaptic connections are seen in ChIs (*Chuhma et al., 2014*). This has revealed subregional heterogeneity in the dStr due to a slow mode of DA neuron glutamate cotransmission in ldStr ChIs, mediated by substantia nigra DA neurons.

## Results

### Different responses of ChIs to DA neuron terminal stimulation in the mdStr and ldStr

To activate DA neuron terminals impinging on recorded Str neurons, we used mice with DA transporter (DAT) driven channelrhodopsin 2 (ChR2)-enhanced yellow fluorescent protein (EYFP) expression (DAT$^{IREScre}$;ChR2-EYFP mice), in which ChR2-EYFP is expressed almost exclusively in ventral midbrain DA neurons (over 98%) (*Mingote et al., 2017*), and wide-field photostimulation. DA neuron synaptic responses were recorded from ChIs in the mdStr or ldStr (*Figure 1A*), identified by soma size and membrane properties (viz. *Chuhma et al., 2014*) (*Figure 1—figure supplement 1*). With train photostimulation of DA neuron terminals, mimicking the phasic firing of DA neurons (five 5-msec pulses at 20 Hz), mdStr ChIs showed the previously reported pause in firing (*Chuhma et al., 2014*; *Straub et al., 2014*), while ldStr ChIs showed a shorter pause in firing followed by an increase in firing (*Figure 1B*). Average ChI firing frequency (in 100 ms bins) was reduced similarly during photostimulation in mdStr and ldStr, while post-stimulation responses differed; firing was reduced in the mdStr ChIs and increased in the ldStr ChIs (*Figure 1C*). To evaluate the changes relative to baseline firing, firing z-scores were calculated. During photostimulation, 0–0.4 s from the onset of the train, firing z-scores in the mdStr (−2.5 ± 0.3) and ldStr ChIs (−1.7 ± 0.3) were both negative, reflecting a reduction in firing, and were not significantly different (p=0.074, independent-sample t-test). During the post-stimulation period, from 0.5 to 0.9 s, firing z-scores in mdStr ChIs remained negative (−2.4 ± 0.4), while they became positive in ldStr ChIs, reflecting an increase in firing (+2.5 ± 0.8) (*Figure 1D*).

Basic membrane properties did not differ significantly between ChIs in the mdStr and ldStr, including baseline firing frequency (mdStr 3.2 ± 0.4 Hz, ldStr: 3.1 ± 0.4 Hz), resting membrane potential(mdStr −62.4 ± 2.0 mV, ldStr −64.6 ± 1.5 mV), action potential threshold (mdStr −50.9 ± 1.8 mV, ldStr −51.8 ± 1.0 mV) and input impedance (mdStr 149.9 ± 15.3 MΩ, ldStr 179.4 ± 18.6 MΩ) (*Figure 1—figure supplement 1*). Modulation of firing of ldStr ChIs was not observed in mice expressing only EYFP in DA neurons (DAT$^{IREScre}$;R26-stop-EYFP), indicating that the responses in ldStr ChIs were not due to blue light illumination or the fluorescent reporter (*Figure 1—figure supplement 2*).

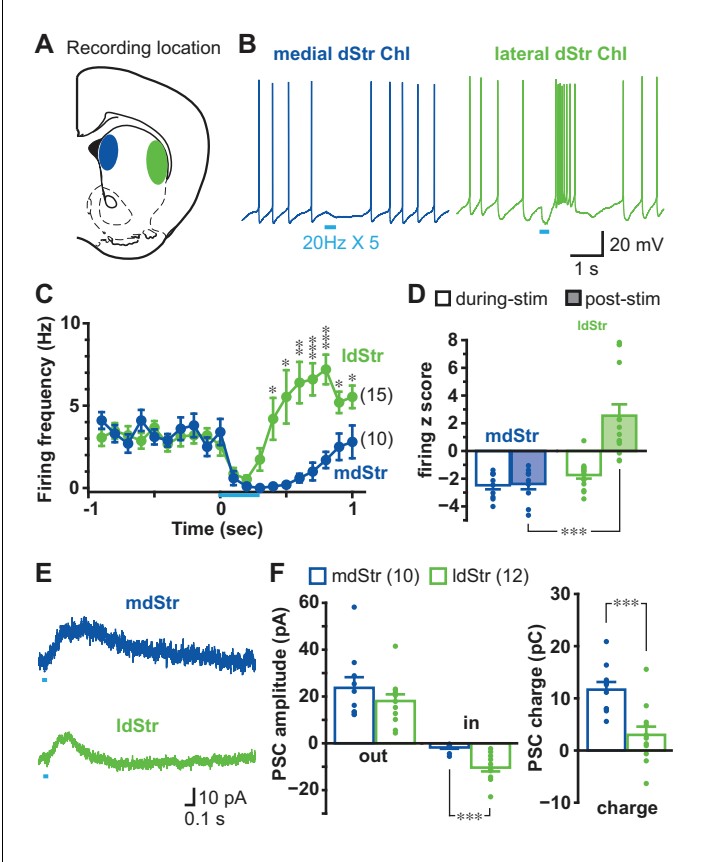

**Figure 1.** Different responses of ChIs to DA neuron terminal stimulation in the mdStr and ldStr. (**A**) DA neuron terminals were activated optogenetically, and recordings made from ChIs in the mdStr (blue) or ldStr (green). (**B**) Train photostimulation (five pulses, 20 Hz; light blue bars) paused ChI firing in the mdStr, while it paused and then increased ChI firing in the ldStr. (**C**) Peristimulus firing frequencies were calculated from 10 consecutive traces (100 msec bins) for each cell, and cells pooled. mdStr and ldStr responses were significantly different (F = 8.1, p=0.000, mixed ANOVA). mdStr: n = 10 cells from nine animals. ldStr: n = 15 cells from nine animals. (**D**) Firing z score during photostimulation (0–0.4 s from train onset; open bars) and post (0.5–0.9 s; shaded bars), for the same cells. F = 22.9, p=0.000, mixed ANOVA; post-hoc regional comparison during-stim p=0.074, post-stim p=0.000. (**E**) Under voltage clamp, PSCs were evoked by single-pulse photostimulation, delivered at 0.1 Hz. Traces are the average of 10 consecutive traces. (**F**) Average amplitude of IPSCs (outward current), EPSCs (inward current), and synaptic response charge transfer during and following stimulation (1 s window). IPSCs showed no regional difference in amplitude (p=0.29, independent sample t-test); EPSCs showed a regional difference in amplitude (p=0.000, Welch's t-test) and charge (p=0.001, independent sample t-test). mdStr: n = 10 cells from six animals. ldStr: n = 12 cells from eight animals. Dots in bar graphs show the average measurement from each recorded cell. Numbers of cells recorded are in parentheses. *, ** and *** indicate p<0.05, p<0.01 and p<0.001, respectively. See also *Figure 1—figure supplements 1*, *2* and *3*.

DOI: https://doi.org/10.7554/eLife.39786.002

The following figure supplements are available for figure 1:

**Figure supplement 1.** Electrophysiological properties of ChIs in the mdStr and ldStr.
DOI: https://doi.org/10.7554/eLife.39786.003
**Figure supplement 2.** Effects of train photostimulation on firing were dependent on ChR2 expression.
DOI: https://doi.org/10.7554/eLife.39786.004
**Figure supplement 3.** PSCs elicited by single pulse photostimulation were dependent on ChR2 expression.
DOI: https://doi.org/10.7554/eLife.39786.005

In voltage clamp recordings done at −70 mV, single-pulse photostimulation of DA neuron terminals (at 0.1 Hz) evoked sub-second PSCs in both mdStr and ldStr ChIs (*Figure 1E*). The peak amplitudes of IPSCs (outward currents) did not differ significantly between mdStr ChIs (23.7 ± 4.5 pA) and

ldStr ChIs (18.1 ± 2.8 pA). Peak amplitude of slow EPSCs (inward currents) was significantly greater in ldStr ChIs (mdStr 1.8 ± 0.6 pA, ldStr 10.4 ± 1.6 pA) (*Figure 1F*). The charge transfer of PSCs (time integration of PCSs) in mdStr ChIs (11.7 ± 1.5 pC) was significantly larger than in ldStr ChIs (3.0 ± 1.6 pC) (*Figure 1F*). PSCs were not seen in ChIs of mice with conditional EYFP expression (*Figure 1—figure supplement 3*). These observations suggest that the shorter firing pause in ldStr ChIs was likely due to shorter IPSCs with the same peak amplitude, which were captured more effectively by measuring charge transfer.

## Synaptic currents underlying different ChI responses in the mdStr and ldStr

Differences in responses in mdStr and ldStr ChIs could be due to differences in DA D2 receptor-mediated IPSCs (*Chuhma et al., 2014*; *Straub et al., 2014*). Application of the D2-antagonist sulpiride (10 µM) blocked IPSCs in both mdStr and ldStr ChIs (mdStr pre-drug 20.8 ± 2.9 pA, sulpiride 2.5 ± 0.6 pA; ldStr pre-drug 18.0 ± 3.8 pA, sulpiride 3.7 ± 0.6 pA), confirming D2R-mediation of the IPSCs (*Figure 2A and B*). After sulpiride application, slow EPSCs became recognizable in both mdStr and ldStr ChIs (mdStr pre-drug 1.5 ± 0.6 pA, sulpiride 4.8 ± 0.7 pA; ldStr pre-drug 9.9 ± 1.5 pA, sulpiride 17.0 ± 2.4 pA), but were more prominent in the ldStr (*Figure 2B*). To evaluate the D2R-mediated component, post-sulpiride traces were subtracted from pre-drug traces to reveal sulpiride-sensitive currents (*Figure 2A*). While pre-drug PSC charge transfer showed significant subregional differences (mdStr 15.3 ± 2.1 pC, ldStr 1.6 ± 3.2 pC), D2-mediated, sulpiride-sensitive components did not (mdStr 17.2 ± 2.1 pC, ldStr 15.8 ± 2.4 pC) (*Figure 2C*). DA release in the mdStr and ldStr measured by fast-scan cyclic voltammetry (FSCV) (*Figure 2D*) did not differ significantly with single-pulse photostimulation (mdStr 0.49 ± 0.07 µM, ldStr 0.51 ± 0.07 µM), with train-pulse stimulation (mdStr 0.66 ± 0.09 µM, ldStr 0.64 ± 0.13 µM) (*Figure 2E*), decay time constant of single-stimulation responses (mdStr 0.29 ± 0.04 s, ldStr 0.31 ± 0.05 s), or of train-stimulation responses (mdStr 0.33 ± 0.03 s, ldStr 0.36 ± 0.05 s) (*Figure 2F*), or the train/single ratio (mdStr 1.36 ± 0.09, ldStr 1.23 ± 0.09) (*Figure 2G*). Thus, the regional differences of PSCs were not due to differences in D2R responsivity or DA release, but rather to a slow EPSC revealed by sulpiride application.

In ldStr ChIs, slow EPSCs were not blocked by a cocktail of ionotropic glutamate receptor (iGluR) antagonists (40 µM CNQX and 100 µM APV), while small fast EPSCs were blocked (slow EPSCs pre-drug 14.2 ± 1.8 pA, CNQX + APV 14.3 ± 2.5 pA; fast EPSCs pre-drug 8.9 ± 2.4 pA, CNQX + APV 1.2 ± 0.1 pA) (*Figure 2H,I*). After blockade of D2Rs and iGluRs, small fast IPSCs were revealed in some cells, which were blocked by the GABA$_A$ antagonist SR95531 (gabazine, 10 µM) (*Figure 2H*). These GABA$_A$ responses, likely due to DA neuron GABA cotransmission (*Straub et al., 2014*), were not prominent when holding at −70 mV, close to the Cl$^-$ reversal potential. When all known DA neuron neurotransmitter receptors were blocked by a cocktail of D2, iGluR and GABA$_A$ antagonists, slow EPSCs were isolated (*Figure 2J*). Firing z-scores during the post-stimulation period stayed positive with the cocktail of antagonists (ctrl 2.5 ± 0.8, antagonists 3.9 ± 0.8), while z-scores during stimulation were close to zero (ctrl −1.7 ± 0.3, antagonists 0.7 ± 0.2) (*Figure 2K*). These slow EPSCs, previously reported by Straub et al. (*Straub et al., 2014*), mediated by an unidentified mechanism, were responsible for the delayed increase of firing in ldStr ChIs with train stimulation (*Figure 2J*). Thus, regional differences in DA neuron synaptic actions between the mdStr and ldStr were due to slow EPSCs in ldStr ChIs, mediated by unknown receptors.

## Cell type and regional distribution of slow EPSCs

To examine the distribution of slow EPSCs across cell types and locations, we recorded from ChIs and the two classes of spiny projection neurons (SPNs) in the mdStr and ldStr. SPNs, which are the principal Str neurons, were identified by either D1-tdTomato (direct-pathway SPNs; dSPNs) or D2-EGFP (indirect pathway SPNs; iSPNs) fluorescence in triple mutant mice, produced by breeding the reporter lines with DAT$^{IREScre}$;ChR2-EYFP mice. Slow EPSCs were evoked by single pulse photostimulation at 0.1 Hz, pharmacologically isolated with a cocktail of D2, iGluR and GABA$_A$ antagonists, and charge transfer measured in the window from 0.2 to 1.7 s after the onset of photostimulation, corresponding to the duration of the EPSC.

Single photostimulation evoked small or no slow EPSCs in mdStr ChIs (2.1 ± 0.5 pC), prominent slow EPSCs in the ldStr ChIs (25.8 ± 4.4 pC), and no PSCs in SPNs in either the mdStr or ldStr (mdStr

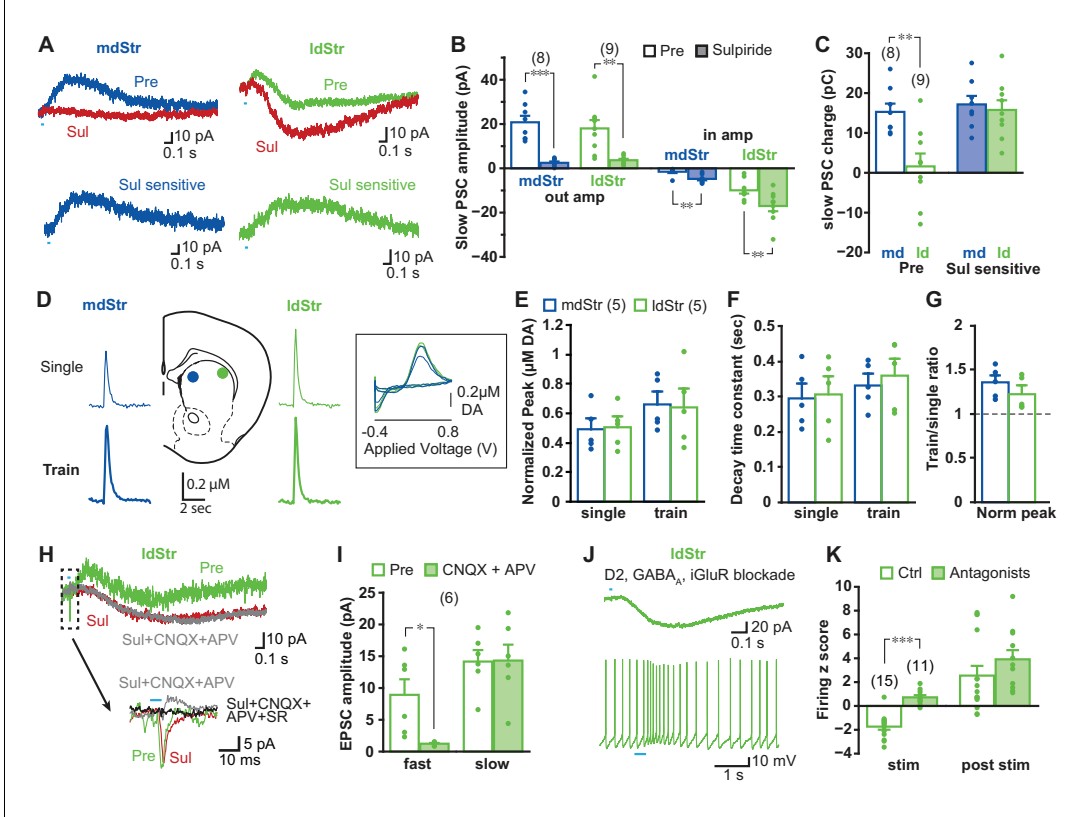

**Figure 2.** Synaptic currents underlying different ChI responses in the mdStr and ldStr. (A) PSCs elicited by single pulse photostimulation in the mdStr (blue) and ldStr (green) before (Pre) and after sulpiride application (Sul, red). Subtraction of traces (bottom) revealed the sulpiride-sensitive D2R PSC. (B) Average amplitudes of IPSCs and EPSCs, before and after sulpiride. Comparison between pre and post sulpiride: mdStr IPSC, p=0.000; ldStr IPSC, p=0.003; mdStr EPSC, p=0.004; ldStr EPSC, p=0.006 (paired t-test). mdStr: n = 8 cells from five animals, ldStr: n = 9 cells from six animals. (C) Average PSC charge transfer in the 1.5 s post-stimulus window before sulpiride application, and the sulpiride-sensitive component in the same cells. Comparison between mdStr and ldStr: charge, p=0.003; sulpiride-sensitive component, p=0.67 (Welch's t-test). (D) DA release was measured with FSCV, in response to single pulse (top, thin traces) and train (bottom, thick traces) photostimulation in the two subregions. Representative cyclic voltammograms are shown in the inset. (E–G) Normalized peak amplitude, decay time constant, and the ratio of train to single stimulation of DA release are plotted. Dots are measurements in single slices. Comparison between mdStr and ldStr: single peak, p=0.83; train peak, p=0.79; single tau, p=0.81, train tau p=0.37, train/single ratio p=0.33, paired t-test. n = 5 slices from five animals. (H) In the ldStr (green), after blocking the D2R component of the slow response with sulpiride (red), CNQX + APV (gray) had no impact on the slow response, but blocked the initial fast inward iGluR component (traces in dashed box are shown on an expanded time scale below); SR69951 blocked the remaining initial fast outward GABA_A component (black trace). n = 6 cells from three animals. (I) Average amplitudes of fast and slow EPSCs before and after CNQX + APV, in the continuous presence of sulpiride. Comparison between pre and post iGluR antagonists: fast EPSC, p=0.022; slow EPSC, p=0.90, paired t-test. (J) Isolation of slow EPSC in ldStr ChIs after blockade of D2, GABA_A and iGlu receptors, with single pulse photostimulation (at 0.1 Hz; above), and effect of train photostimulation on firing (5 pulses at 20 Hz; bottom). (K) Firing z score during and after train photostimulation, prior to (Ctrl) and after application of a cocktail of D2 + GABA_A + iGluR antagonists, recorded from different sets of neurons. Ctrl data are the same as shown in *Figure 1C*. Ctrl: n = 15 cells from nine animals. Antagonists: n = 11 cells from seven animals. Comparison between pre and post antagonists: during stim p=0.000, post stim p=0.25, independent sample t-test. Numbers in parentheses in B, C, I, K indicate numbers of cells recorded and in E indicate numbers of slices recorded. Dots in bar graphs show the average for each recorded cell. Sample PSC traces are the average of 10 consecutive traces. *, ** and *** indicate p<0.05, p<0.01 and p<0.001, respectively.

DOI: https://doi.org/10.7554/eLife.39786.006

SPN 0.93 ± 0.24 pC, ldStr SPN 0.64 ± 0.15 pC) (*Figure 3A, B and C*). The charge transfer of slow EPSCs ranged from 0.014 to 6.7 pC in mdStr ChIs and from 3.9 to 140.0 pC in ldStr ChIs (*Figure 3B and C*). Charge transfer was significantly different between cell types, between regions, with a significant cell type by region interaction. ldStr ChIs showed significantly larger responses than SPNs or mdStr ChIs. mdStr ChIs showed significantly larger responses than SPNs, in either the mdStr or ldStr. No significant difference was observed between mdStr SPNs and ldStr SPNs. When SPNs were split into dSPNs and iSPNs, there were no differences in charge transfer between region or cell type

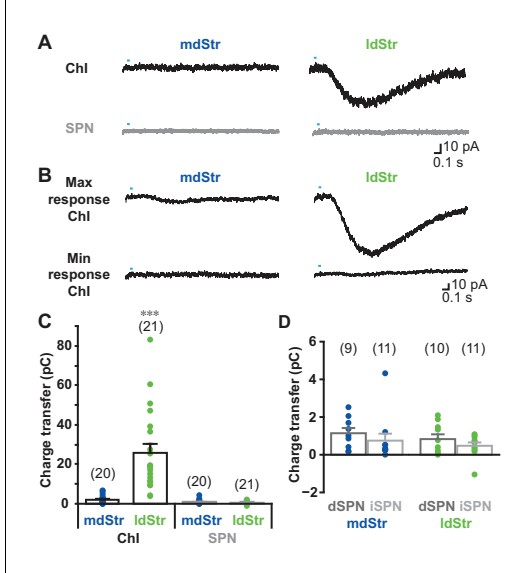

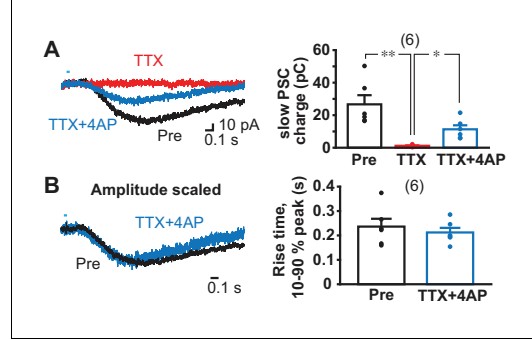

**Figure 3.** Cell type and regional distribution of slow EPSCs. (A) Slow EPSCs were isolated with a cocktail of D2 + GABA$_A$ + iGluR antagonists, and recorded in ChIs and SPNs in the two dStr regions. Traces shown are the average of 10 consecutive responses (at 0.1 Hz). (B) The maximum and minimum responses are shown. (C) Slow EPSCs were measured as the charge transfer in a 1.5 s post-photostimulation time window. Cell type: F = 32.9, p=0.000; region: F = 26.0, p=0.000; cell type/region interaction: F = 27.3, p=0.000, two-way ANOVA. See *Figure 3—source data 1* for result of post-hoc tests. ***p<0.001, ldStr ChIs compared to mdStr ChIs, mdStr SPNs or ldStr SPNs. mdStr ChI: n = 20 cells from eight animals. ldStr ChI: n = 21 cells from 10 animals. mdStr SPN: 20 cells from nine animals. ldStr SPN: 21 cells from nine animals. (D) SPNs responses shown in C are split into dSPNs and iSPNs. SPN type: F = 1.76, p=0.19; region: F = 1.08, p=0.31; SPN type/region interaction: F = 0.005, p=0.95, two-way ANOVA. mdStr dSPN: 9 cells from four animals. mdStr iSPN: 11 cells from five animals. ldStr dSPN: 10 cells from four animals. ldStr iSPN: 11 cells from five animals. Numbers of cells recorded are in parentheses. Dots show the average charge transfer for each cell recorded.

DOI: https://doi.org/10.7554/eLife.39786.007

The following source data is available for figure 3:

**Source data 1.** (statistics for *Figure 3C*).
DOI: https://doi.org/10.7554/eLife.39786.008

**Figure 4.** Monosynapticity of slow EPSCs. (A) Monosynapticity of the slow EPSC was tested by TTX application (1 μM, red) followed by 4-AP (0.5 mM, blue). F = 17.3, p=0.001, one-way repeated-measures ANOVA; post-hoc comparison Pre/TTX p=0.007, TTX/TTX + 4 AP p=0.011, Pre/TTX + 4 AP p=0.014. * and ** indicate p<0.05 and p<0.01, respectively. n = 6 cells from four animals. (B) Scaling traces shown in A to compare EPSC rise time prior to and following TTX + 4 AP (p=0.29, paired t-test), indicating that the second slower component due to polysynaptic transmission was less likely. Dots show the average charge transfer for each recorded cell. Numbers of cells recorded are in parentheses. See also *Figure 4—figure supplement 1*.

DOI: https://doi.org/10.7554/eLife.39786.009

The following figure supplement is available for figure 4:

**Figure supplement 1.** Slow EPSC is mediated by ventral midbrain DA neurons.
DOI: https://doi.org/10.7554/eLife.39786.010

(*Figure 3D*); slow EPSCs were not seen in either dSPNs or iSPNs. Thus, slow EPSCs were exclusive to ChIs and prominent in ldStr ChIs.

## Pharmacological analysis of slow EPSCs

To test monosynapticity of slow EPSCs in ldStr ChIs, DA neuron terminals were stimulated after application of 1 μM tetrodotoxin (TTX) and 0.5 mM 4-aminopyridine (4AP) (*Figure 4*). In this condition, action potentials are blocked by TTX, and axon terminals slightly depolarized by 4AP, but not enough to cause transmitter release; optical stimulation causes further depolarization only in ChR2-expressing terminals and transmitter release (*Petreanu et al., 2009*). Thus, monosynaptic connections generate postsynaptic responses, while polysynaptic connections fail. Application of TTX blocked slow EPSCs completely (pre-drug 26.7 ± 5.6 pC, TTX 1.2 ± 0.3 pC), confirming the slow EPSCs were action-potential dependent (*Figure 4A*). Addition of 4AP partially recovered slow EPSCs (11.3 ± 2.5 pC), indicating that slow EPSCs have a monosynaptic component (*Figure 4A*). To determine whether there was an additional polysynaptic component, the rising phase of EPSCs was compared before and after TTX + 4 AP (*Figure 4B*). Traces were superimposable after amplitude scaling; the rise time (10% to 90% of peak

amplitude) was not different (pre-drug 0.24 ± 0.03 s, TTX + 4 AP 0.21 ± 0.02 s), indicating that the slow EPSCs are solely monosynaptic.

To confirm that slow EPSCs arose from ventral midbrain DA neurons, we injected a conditional viral vector expressing ChR2 (AAV-DIO-ChR2-EYFP) into the ventral midbrain of DAT[IRIScre] mice and found that the responses did not differ from those recorded in DAT[IRIScre];ChR2-EYFP mice (*Figure 4—figure supplement 1*).

When G-protein coupled receptor (GPCR) transduction was blocked by GDPβS (0.5–1 mM) in the intracellular solution, slow EPSCs were almost completely blocked 9 min after entering whole cell mode, while EPSCs were not significantly reduced with control intracellular solution (with GTP), showing that the blockade was not due to cell deterioration (comparison to 0–1 min from achieving whole cell, Ctrl 5–6 min 115.2 ± 16.1%, 9–10 min 97.0 ± 15.7%; GDPβS 5–6 min 13.7 ± 8.0%, 9–10 min 6.0 ± 2.9%) (*Figure 5A*). To confirm complete blockade of the slow EPSCs, we applied train stimulations after 10 min recording with single stimulation (*Figure 5—figure supplement 1*). The slow EPSCs were not observed in GDPβS dialyzed cells (2.5 ± 0.9 pC), while slow EPSCs persisted in control cells (36.1 ± 10.2 pC). Thus, slow EPSCs were GPCR mediated.

Next, we sought to determine the receptors mediating the slow EPSC, which should be (1) GPCRs, (2) capable of exciting neurons within a second, (3) expressed in ChIs, and (4) responsive to neurotransmitters released by DA neurons. Two receptor types fulfill these criteria: DA D1-like (D1/D5) receptors and group I metabotropic glutamate receptors (mGluRs). D1-like receptors mediate a delayed firing increase in olfactory tubercle ChIs (*Wieland et al., 2014*), and ldStr ChIs are likely to share the same mechanisms. mGluR1s mediate slow EPSCs in cerebellar Purkinje cells, which have a time course similar to the slow EPSCs in ldStr ChIs (*Batchelor and Garthwaite, 1997*; *Tempia et al., 1998*). Bath application of group 1 mGluR agonist elicits inward currents in dStr ChIs (*Berg et al., 2007*; *Takeshita et al., 1996*) sufficient to mediate slow EPSCs. Thus, we examined the effects of D1- and group I mGluR (mGluR1/5) antagonists on slow EPSCs.

We used a cocktail of a novel selective and potent mGluR1 antagonist JNJ16259685 (JNJ; 10 µM) and a selective and potent mGluR5 antagonist MTEP (10 µM) to block group I mGluRs. Bath application of the D1 antagonist (antagonist of D1-like receptors) SCH23390 (SCH; 10 µM) or a cocktail of JNJ + MTEP significantly reduced slow EPSCs after 20 min, compared to control (Ctrl) (% pre-drug; Ctrl 95.1 ± 5.4%, SCH 61.4 ± 9.5%, JNJ + MTEP 36.1 ± 5.2%) (*Figure 5B and C*). While JNJ + MTEP showed significant effects after 10 min (Ctrl 103.3 ± 2.9%, JNJ + MTEP 50.2 ± 5.3%), SCH did not (SCH 78.7 ± 8.6%), suggesting that longer application was required for SCH action. While JNJ + MTEP effects could not be reversed (44.5 ± 12%), as shown in cerebellum slices (*Fukunaga et al., 2007*), the slow EPSC reversed to almost pre-drug levels by wash after SCH (95.5 ± 3.8%). The combined application of SCH + JNJ + MTEP almost completely eliminated the slow EPSC after 20 min (10.1 ± 1.8%) (*Figure 5B and C*), indicating that slow EPSCs were mediated by mGluR1/5 and D1/5R, but mainly by mGluRs. The slow EPSC partially reversed after SCH + JNJ + MTEP (37.6 ± 5.4%), as only the SCH effects apparently reversed. With JNJ alone, the inhibition after 20 min application did not differ from JNJ + MTEP (JNJ 32.8 ± 6.8% pre-drug), while MTEP alone had no significant effect (MTEP 100.6 ± 7.0% pre-drug) (*Figure 5D*), indicating that the group I mGluR component of the slow EPSC was mGluR1 mediated.

Bath application of mGluR group I agonists have been shown to depolarize dStr ChIs by activation of transient receptor potential channel (TrpC) 3 and 7 (*Berg et al., 2007*), pointing to TrpC mediation of the slow EPSC. We blocked D2 and GABA$_A$ receptors, but not iGluRs to show that antagonist actions were selective for the slow EPSC, as TrpC antagonists may affect transmitter release by reducing presynaptic excitability (*Yau et al., 2010*). Bath application of the TrpC3 selective antagonist Pyr3 (20 µM) reduced the slow EPSC to 27.2 ± 5.1% pre-drug, without affecting the fast EPSC (89.8 ± 12.2%) (*Figure 6A and B*). The TrpC3/7 antagonist flufenamic acid (FFA, 100 µM) blocked the slow EPSC completely (3.9 ± 1.1% pre-drug), without affecting the fast EPSC (89.5 ± 4.2% pre-drug) (*Figure 6A and B*), indicating that the slow EPSCs were mainly TrpC3 mediated. Taken together, these results show that slow EPSCs were mediated by mGluR1 and D1/5R through TrpC3/7 as the effector channel, but mainly by mGluR1 through TrpC3.

## DA neuron glutamate cotransmission in the lateral dStr

While DA neuron glutamate signals are lacking in the mdStr (*Mingote et al., 2015*), the mGluR1 component of the slow EPSC in ChIs in the ldStr indicates that DA neuron glutamate cotransmission

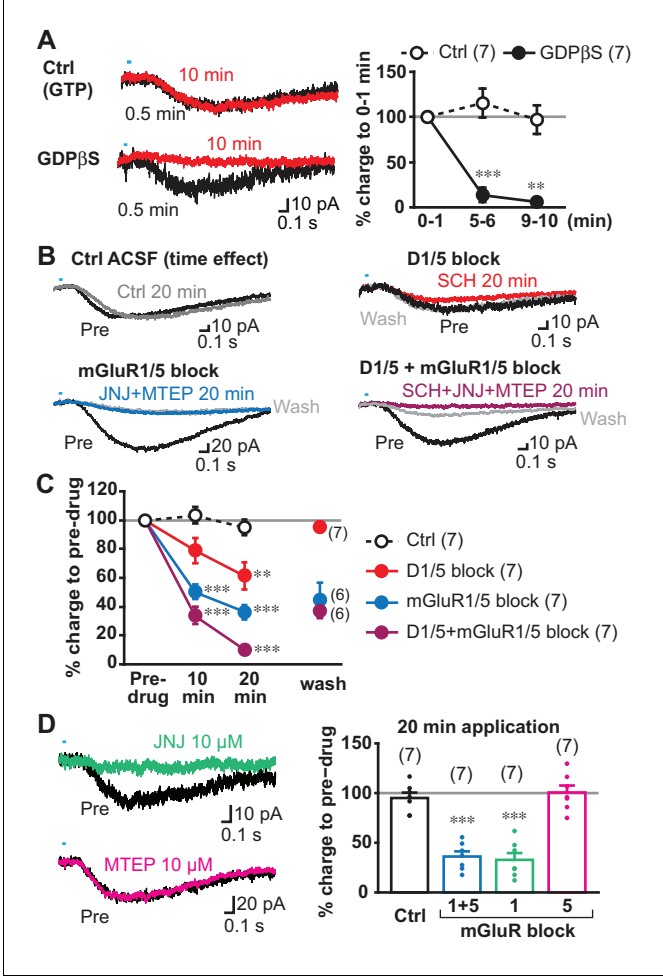

**Figure 5.** Transmitter receptors mediating slow EPSCs. (**A**) G-protein mediation was tested by comparing recordings with GDPβS or GTP (Ctrl) in the intracellular solution. Sample traces are shown (left) for 0–1 min after achieving whole cell mode (average from six traces, 0.5 min; black), and then after 9–10 min after (10 min; red), allowing time for intracellular dialysis. Time course of GDPβS action is shown (right), with charge transfer normalized to the initial responses (gray line). Time/treatment interaction: F = 24.4, p=0.000 (mixed ANOVA); post-hoc comparison between subject: 5 min p=0.000, 10 min p=0.001. Ctrl: n = 7 cells from four animals, GDPβS: n = 7 cells from four animals. ** and *** indicate p<0.01 and p<0.001, respectively, compared to Ctrl. (**B**) Pharmacological identification of GPCRs mediating the slow EPSC. Sample traces for pre-drug application (black), and 20 min after continued perfusion of ACSF (Ctrl; top left, gray), SCH23390 (SCH, 10 µM; top right, red), a cocktail of JNJ16259685 (JNJ, 10 µM)+MTEP (10 µM; bottom left, blue) or SCH + JNJ + MTEP (bottom right, purple), and after wash (light gray). (**C**) Time course of antagonist actions on EPSC charge transfer are shown, followed by a wash, normalized to pre-drug control. The number of cells recorded are given in the legend; the numbers of cells with wash data are indicated on the graph. Time/treatment F = 21.4, p=0.000, mixed ANOVA. See *Figure 5—source data 2* for results of post-hoc tests. Ctrl: n = 7 cells from four animals. SCH: n = 7 cells from five animals. JNJ + MTEP: n = 7 cells from four animals. SCH + JNJ + MTEP: n = 7 cells from four animals. ** and *** indicate p<0.01 and p<0.001, respectively, compared to Ctrl. (**D**) Identification of mGluR Group I receptor subtype. The mGluR1 antagonist JNJ significantly attenuated the slow EPSC, after 20 min (left top, green), while the mGluR5 antagonist MTEP had no effect (left bottom, magenta). Percent reduction in charge transfer is shown for JNJ (mGluR1 blockade, green) and MTEP (mGluR5 blockade, red). For comparison (right), Ctrl and mGluR1/5 20 min data points in C are replotted (white and blue bars, respectively). Comparison to pre-drug (100%) at 20 min: Ctrl p=0.40, mGluR1/5 antagonists p=0.000, mGluR1 antagonist p=0.000, mGluR5 antagonist p=0.93, one-sample t-test. Comparison among treatment at 20 min: F = 35.3, p=0.000, one-way ANOVA. See *Figure 5— source data 2* for results of post-hoc tests. JNJ: n = 7 cells from four animals. MTEP: n = 7 cells from five animals. *** indicates p<0.001 compared to 100% level. Dots show the average charge transfer for each recorded cell.

*Figure 5 continued on next page*

*Figure 5 continued*

Numbers of cells recorded are in parentheses. See also *Figure 5—figure supplement 1* and *Figure 5—source data 1*.

DOI: https://doi.org/10.7554/eLife.39786.011

The following source data and figure supplement are available for figure 5:

**Source data 1.** (individual cell data for line graphs in *Figure 5A and C*)

DOI: https://doi.org/10.7554/eLife.39786.013

**Source data 2.** (statistics for *Figure 5C and D*).

DOI: https://doi.org/10.7554/eLife.39786.014

**Figure supplement 1.** The slow EPSC is blocked completely after 10 min intracellular dialysis of GDPβS.

DOI: https://doi.org/10.7554/eLife.39786.012

extends beyond the ventral Str. If so, other cell types in the ldStr may show glutamate EPSCs. Indeed, fast glutamatergic EPSCs have been variably reported in SPNs in the Str, but without specification of subregion (*Stuber et al., 2010*; *Tritsch et al., 2012*). To examine DA neuron glutamate EPSCs in SPNs, we used a glutamate-isolation cocktail to block GABA, DA and cholinergic receptors (SR95531 10 µM, CGP54626 3 µM, SCH23390 10 µM, sulpiride 10 µM, scopolamine 2 µM, mecamylamine 10 µM). EPSCs were recorded at −70 mV with a Cs⁺-based pipette solution to improve space clamp, and QX-314 (5 mM) to block unclamped action potentials. Single pulse stimulation at 0.1 Hz evoked fast EPSCs in ldStr dSPNs (86.9 ± 12.3 pA) and iSPNs (30.9 ± 5.4 pA), while only small EPSCs were observed in ldStr ChIs (5.9 ± 2.0 pA) (*Figure 7A and B*). Since membrane properties could not be examined with the Cs⁺ pipette solution, we used triple mutant DAT^IREScre;ChR2-EYFP;ChAT-GFP mice for ChI recordings. In ldStr ChIs, slow EPSCs were also observed, presumably mediated by mGluR1 because D1-like receptors were blocked by the glutamate-isolation cocktail (*Figure 7A*, arrows). Fast EPSCs were rarely observed in the mdStr regardless of postsynaptic cell type (dSPN 6.5 ± 2.0 pA, iSPN 3.0 ± 0.4 pA, ChI 8.8 ± 4.3 pA) (*Figure 7A and B*). Fast EPSCs in the dSPNs and iSPNs were significantly larger in the ldStr than mdStr, while regional differences were not observed in ChIs. The fast EPSCs in ldStr SPNs were blocked by CNQX + APV, confirming mediation by iGluRs (5.7 ± 1.2% pre-drug) (*Figure 7A and C*).

To confirm that glutamate responses originated from DA neurons, we examined responses in vesicular glutamate transporter 2 (VGLUT2) conditional knockout mice (cKO; DAT^IREScre;ChR2-EYFP; VGLUT2^lox/lox) and controls (DAT^IREScre;ChR2-EYFP; VGLUT2^+/+). Fast EPSCs were absent in ldStr SPNs in cKOs, but not in controls (cKO 1.1 ± 0.3 pA, Ctrl 26.1 ± 6.0 pA) (*Figure 7D*). Slow EPSCs in ldStr ChIs isolated by a cocktail of D2, GABA_A and iGluR antagonists were significantly smaller in VGLUT2 cKO mice, although the responses were not completely eliminated (cKO 3.0 ± 0.8 pC, Ctrl 13.7 ± 2.6 pC) (*Figure 7E*). When the D1-antagonist SCH23390 was added to the cocktail of antagonists, slow EPSCs were almost completely abolished, confirming glutamate and D1 mediation (cKO 1.3 ± 0.3 pC, Ctrl 10.2 ± 2.4 pC) (*Figure 7F*). Thus, in the ldStr, DA neuron glutamate cotransmission engages different glutamate receptors in different postsynaptic target cells — mGluRs mediating slow EPSCs in ChIs and iGluRs mediating fast EPSCs in SPNs; in the mdStr, very little glutamate cotransmission is seen in either ChIs or SPNs.

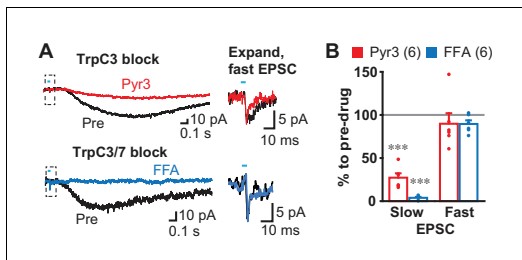

**Figure 6.** Slow EPSC effector channels. (A) Identification of ion channels coupled to G-protein coupled receptors. D2 and GABA_A antagonists were used to isolate fast and slow EPSCs. The TrpC3 selective antagonist Pyr3 (20 µM; left top, red) or TrpC3/7 antagonist FFA (100 µM, left bottom, blue) each significantly attenuated the slow EPSC. Fast EPSCs were unaffected (dashed boxes, left), as shown on an expanded time scale in right. Sample traces are the average of 10 consecutive traces. (B) Summary of drug responses. Comparison to pre-drug (100%): Pyr3 slow p=0.000, FFA slow p=0.000, Pyr3 fast p=0.44, FFA fast p=0.056, one sample t-test. Pyr3: n = 6 cells from four animals. FFA: n = 6 cells from three animals. Dots show the average charge transfer for each recorded cell. Numbers of cells recorded are in parentheses. *** indicates p<0.001 compared to 100% level. A gray line in graph indicates the pre-drug charge transfer (100%).

DOI: https://doi.org/10.7554/eLife.39786.015

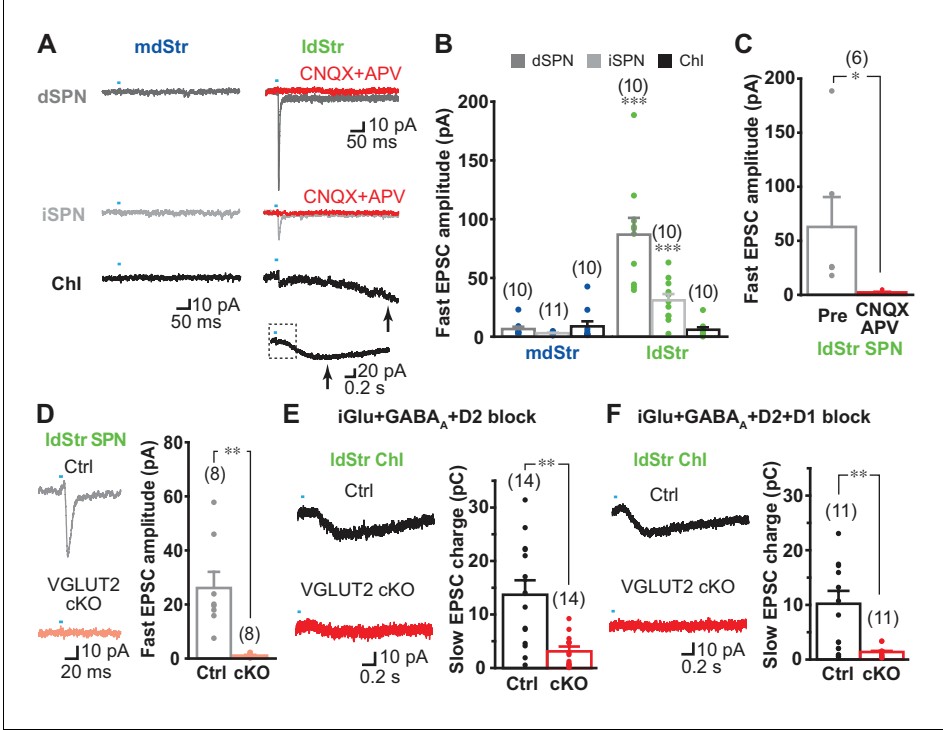

**Figure 7.** DA neuron glutamate cotransmission in the lateral dStr. (**A**) Recording from dSPNs, iSPNs and ChIs in the mdStr and ldStr revealed fast EPSCs in ldStr SPNs that were blocked by CNQX + D APV (red traces). Glutamate EPSCs were pharmacologically isolated. The slow EPSC in the ldStr ChI (arrow) is shown below on a slower time scale (with the time span of the fast trace indicated by a dashed rectangle). (**B**) Mean amplitudes of fast EPSC are shown by cell type and subregion. Cell type: F = 19.4, p=0.000; region: F = 43.4, p=0.000; cell type/region interaction F = 20.4, p=0.000, two-way ANOVA. Post-hoc comparison between regions: dSPN p=0.000, iSPN p=0.000, ChI p=0.56. mdStr dSPN: n = 10 cells from four animals. mdStr iSPN: n = 11 cells from three animals. mdStr ChI: n = 10 cells from three animals. ldStr dSPN: n = 10 cells from four animals. ldStr iSPN: n = 10 cells from three animals. ldStr ChI: n = 10 cells from four animals. *** indicates p<0.001 compared to mdStr. (**C**) Fast EPSCs were blocked completely by CNQX + APV. Comparison to pre-drug (100%): p=0.000, one-sample t-test. n = 6 cells from three animals. (**D–F**) Fast and slow EPSCs in the ldStr recorded from mice with DA neuron selective VGLUT2 KO (cKO). (**D**) Fast EPSCs in ldStr SPNs are shown (left) and the average amplitude (right). Comparison between genotype: p=0.004, Welch's t-test. n = 8 cells from two animals. (**E**) Slow EPSCs in ldStr ChIs isolated by application of iGluR + GABA_A + D2 antagonists (p=0.001, Welch's t-test) and (**F**) after addition of a D1 antagonist (p=0.003, Welch's t-test). Ctrl (**E**): n = 14 cells from three animals. cKO (**E**): n = 14 cells from three animals. Ctrl (**F**): n = 11 cells from three animals. cKO(**F**): n = 11 cells from three animals. Dots show the average charge transfer for each recorded cell and bars the mean and S.E.M. for all recorded cells. Sample traces are the average of 10 consecutive traces. Dots show the average measurement for each recorded cell. Numbers of cells recorded are in parentheses. ** indicates p<0.01 for comparison between genotypes.

DOI: https://doi.org/10.7554/eLife.39786.016

## Substantia nigra DA neurons mediate glutamate cotransmission in the ldStr

DA neurons project to the Str topographically, with the most medially located VTA neurons projecting to the NAc medial shell and more lateral SN DA neurons projecting to the ldStr (*Haber et al., 2000*; *Ikemoto, 2007*). Based on this topography, projections to the ldStr should be from SN DA neurons. However, in adult mice, VGLUT2 expression is prominent in VTA but not SN pars compacta (SNc) DA neurons (*Morales and Root, 2014*), arguing that non-topographical projections of VTA DA neurons mediate glutamate EPSCs in the ldStr. To evaluate the projections of DA neurons to the mdStr or ldStr, we injected the retrograde axonal tracer cholera toxin B subunit (CTB) into the ldStr, and for comparison into the mdStr (*Figure 8A*, top panels). In the ventral midbrain, double immunostaining for CTB (green) and the DA neuron marker tyrosine hydroxylase (TH; magenta) revealed

DA neurons projecting to the injection sites (**Figure 8A**, middle). CTB is transported anterogradely as well, and so revealed the terminals of Str projection neurons in the SN pars reticulata (SNr) (**Figure 8A**, middle, asterisks). The restricted distribution of anterogradely labeled terminals confirmed that the CTB injections were limited to the mdStr or ldStr, and indicated that the wider CTB staining in the Str was not due to the wider spread of the injected CTB, but rather reflected retrograde and anterograde labeling of locally connected Str neurons. We examined the relative distribution of CTB positive (CTB$^+$) Str-projecting neurons in the VTA and the SN (for delineation, see **Figure 8—figure supplement 1**). Since the division of the SNc and SNr was not clear caudally (**Figure 8—figure supplement 1**), SN (SNc/SNr) cell counts are reported. With mdStr injections, 64 ± 12 cells in the VTA and 323.5 ± 86.5 cells in the SN were CTB$^+$ (**Figure 8A**, bottom). Among them, 33 ± 8 cells (50.4 ± 3.2% of CTB$^+$) in the VTA and 213 ± 51 cells (66.4 ± 2.8% of CTB$^+$) in the SN were TH$^+$ (**Figure 8A**, bottom), demonstrating that most DA neurons projecting to the mdStr are in the SN. With ldStr injections, 409.7 ± 116.0 cells in the VTA and 223 ± 133 cells in the SN were CTB$^+$. Of these, 242 ± 152 cells (52.5 ± 19.9% of CTB$^+$) in the VTA and 296 ± 88 cells (61.2 ± 23.1% of CTB$^+$) in the SN were TH$^+$ (**Figure 8A**, bottom), demonstrating that about the same numbers of DA neurons in the VTA and SN project to the ldStr. The distribution of CTB$^+$/TH$^+$ cells in the VTA and SN differed significantly between mdStr and ldStr injections. Thus, there are non-topographical VTA DA neuron projections to the ldStr, but not to the mdStr. However, this did not resolve whether DA neuron glutamate cotransmission arose from non-topographical VTA DA neuron projections.

To address this, we injected the conditional retrograde viral tracer canine adenovirus 2 (CAV2)-FLEX-ZsGreen (**Ekstrand et al., 2014**) into the mdStr or ldStr of VGLUT2$^{IREScre}$ mice to label glutamatergic (VGLUT2$^+$) neurons projecting to the sites injected. With mdStr injections, no ZsGreen$^+$ cells were seen, consistent with the dearth of DA neuron glutamate cotransmission in the mdStr (**Figure 8B**, middle, left panels). With ldStr injections, there were 8 ± 4 ZsGreen$^+$ cells in the VTA and 157 ± 13 in the SN (**Figure 8B**, middle, right panels), demonstrating that glutamatergic neurons projecting to the ldStr are mostly in the SNc. Among them, 2 ± 1 cells (24.6 ± 6.8% of ZsGreen$^+$) in the VTA and 90 ± 18 cells (57.0 ± 8.5%) in the SN were DA neurons (TH$^+$) (**Figure 8B**, bottom). Thus, topographically projecting SN DA neurons mediate glutamate cotransmission in the ldStr. However, variable expression of CAV receptors on DA neuron terminals may have limited the number of DA/glutamate neurons identified.

To visualize VGLUT2-expressing DA neurons in their entirety, we used the INTRSECT strategy (**Fenno et al., 2014**). We injected a cre-on/flp-on ChR2-EYFP virus (AAV-Con/Fon-ChR2-EYFP) into the SN of VGLUT2$^{IREScre}$; TH$^{2A-flpo}$ mice to express ChR2-EYFP conditionally in DA neurons capable of glutamate cotransmission (**Figure 9A,B**). Due to spread of virus, ChR2-EYFP expression was also observed in the VTA (**Figure 9B**). In the SN, there were 247 ± 11 TH$^+$/ChR2$^+$ and 890 ± 116 TH$^+$/ChR2$^-$ neurons (n = 4 animals), corresponding to 20.7% and 74.8% of all immunopositive (TH$^+$ and/or ChR2$^+$) neurons, respectively (**Figures 9B** and **7C**). There were 53 ± 7 TH$^-$/ChR2$^+$ neurons, so the specificity of ChR2 expression in DA neurons was 82.5%. Of TH$^+$ neurons, 22% were ChR2$^+$ indicative of VGLUT2 expression, making them capable of glutamate cotransmission (**Figure 9C**). Co-labeled terminals were denser in the ldStr, than in the mdStr (**Figure 9D**). The observation of sparse fibers in the mdStr, indicated that ZsGreen labeling underestimated the number of DA neurons capable of glutamate cotransmission, because of the small injection volume. Thus, a substantial number of SN DA neurons express VGLUT2 and project preferentially to the ldStr, coincident with the glutamate cotransmission in the ldStr.

## Expression of slow EPSC mediators in ChIs

While topographic DA neuron projections account for the presence of the slow EPSC in the ldStr, they do not account for the specificity for ChIs (viz. **Figure 3**). To identify the receptors and channels mediating slow EPSCs in ChIs, we used ChATcre;RiboTag mice, with conditional expression of a hemagglutinin-tag on the last exon of ribosome protein RPL22, under the control of the choline acetyltransferase (ChAT) promoter, to enable immunoprecipitation (IP) of ribosome-associated mRNA (**Sanz et al., 2009**) from ChIs. We examined expression of the following genes by quantitative PCR (qPCR): mGluR1 and 5, TrpC 3 and 7, as possible determinants of the slow EPSC; D1R, D5R and D2R, as known controls; ChAT and vesicular acetylcholine transporter (VAChT), as IP controls. We used ΔCt normalized to GAPDH for gene expression, and ΔΔCt to whole Str mRNA to evaluate enrichment of expression in ChIs. Confirming the successful isolation of ChI mRNA, both ChAT and

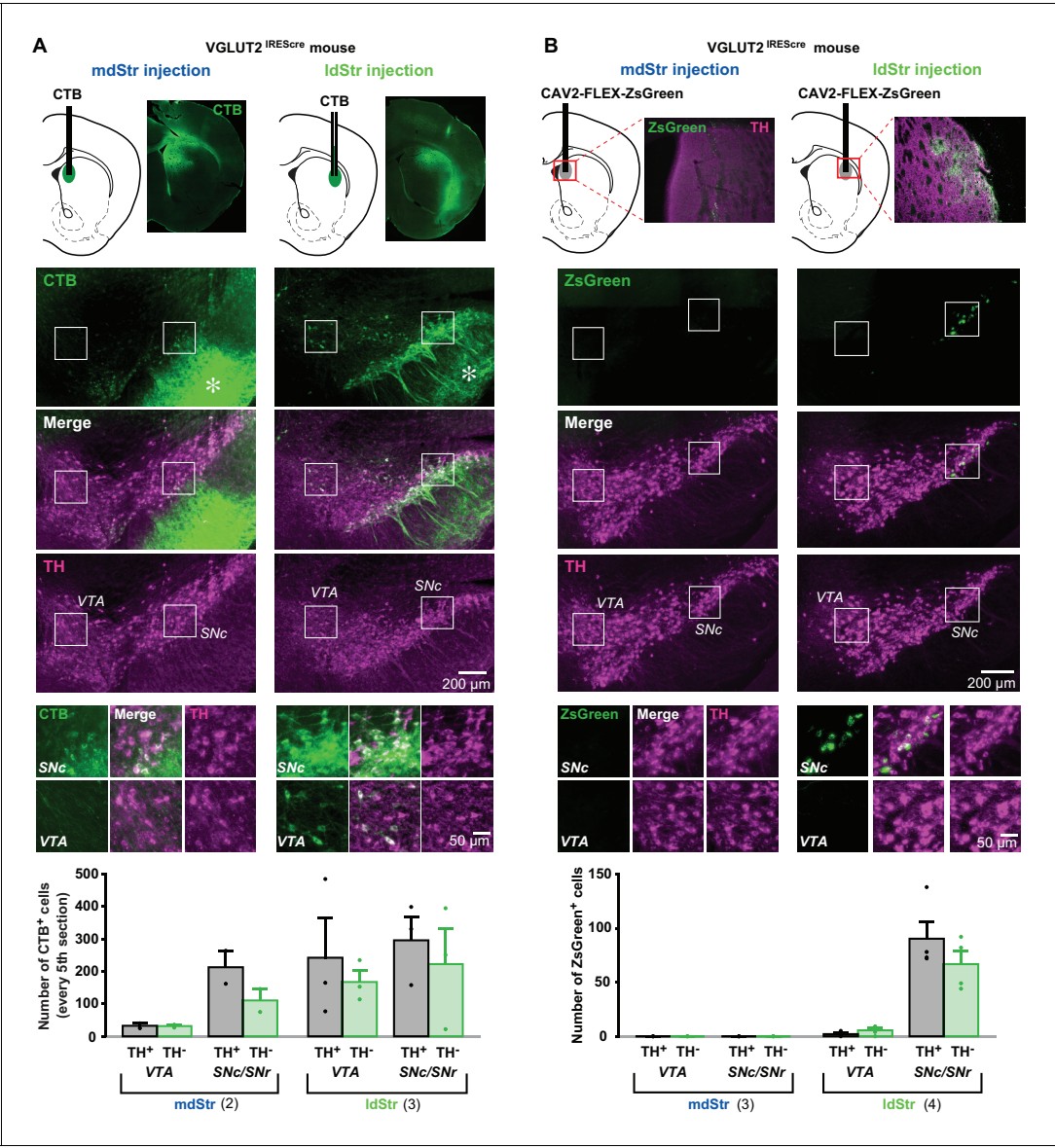

**Figure 8.** Substantia nigra DA neurons mediate glutamate cotransmission in the ldStr. Retrograde tracer injections in the dStr of VGLUT2[IREScre] mice. (A) Top row shows CTB injection sites (schematics) and CTB immunostaining in the Str. The bright green diffuse staining surrounding the injection site reflects CTB uptake by locally connecting Str neurons, which are filled in their entirety. Middle three rows show low magnification images of CTB (green) and TH (magenta) staining in the VTA and SN, with merge images in between. Robust anterograde labeling of Str neuron projections is seen in the SNr (asterisks). White squares outline regions shown at higher magnification in the bottom row. (B) Top row shows CAV2-FLEX-ZsGreen injection sites and ZsGreen fluorescence of transduced VGLUT2[+] neuron terminals in the Str. TH immunostaining (magenta) was done as a counterstain to show Str boundaries. Middle three rows show low magnification images of ZsGreen fluorescence (green) and TH immunostaining in the VTA and SN (magenta), with merge images in between. White squares outline regions shown at higher magnification in the bottom row. All images are in the same orientation as the schematics. Graphs show cell counts in the VTA and SN (SNc and SNr combined) for CTB and CAV2-FLEX-ZsGreen. CTB counts were made from every fifth section; ZsGreen counts were made from all sections. White bars indicate numbers of TH[+]/tracer[+] cells; green bars indicate numbers of TH[−]/tracer[+] cells. Dots show cell counts for each animal. Cell count for CTB: chi-square 163.0, exact significance 0.000 (Pearson's chi-square test). Cell count for ZsGreen: injection location/cell body location interaction, F = 20.2, p=0.006 (mixed ANOVA); post-hoc comparison between injection sites: VTA p=0.16, SN p=0.011. Numbers of animals used are in parentheses. For delineation, see *Figure 8—figure supplement 1*.
DOI: https://doi.org/10.7554/eLife.39786.017

The following figure supplement is available for figure 8:

**Figure supplement 1.** Delineation of ventral midbrain DA neuron groups.
DOI: https://doi.org/10.7554/eLife.39786.018

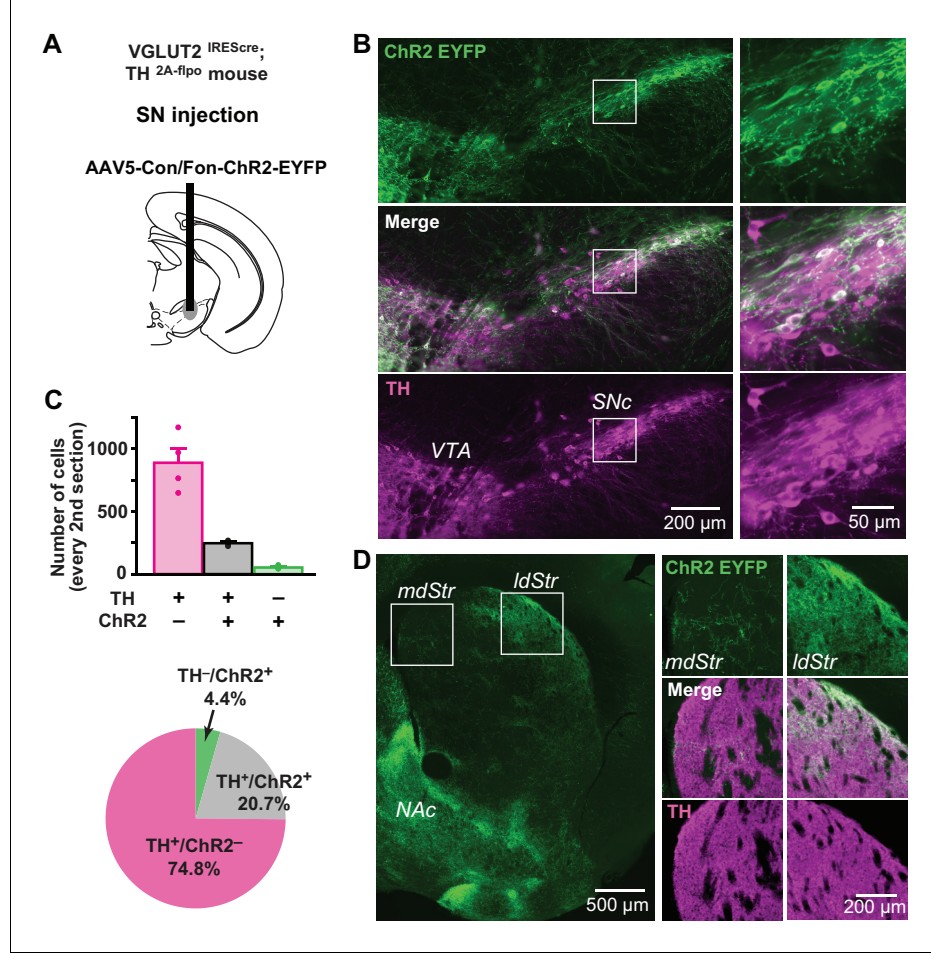

**Figure 9.** Distribution of TH and VGLUT2 coexpressing substantia nigra neurons. (**A**) To visualize projections from TH/VGLUT2 coexpressing neurons, AAV-Con/Fon-ChR2-EYFP was injected in the SN of VGLUT2$^{IREScre}$; TH$^{2A-flpo}$ double hemizygous mice. Dashed lines outline the SN and VTA. (**B**) Low magnification images of ChR2 (green) and TH (magenta) staining in the SN (injection site), with merge images in between. White squares outline regions shown at higher magnification on the right. (**C**) Top panel shows cell counts in the SN for TH$^+$/ChR2$^-$, TH$^+$/ChR2$^+$ and TH$^-$/ChR2$^+$ cells. Cell counts were made from every second section. Dots show cell counts for each animal. Pie chart shows the ratio of TH$^+$/ChR2$^-$, TH$^+$/ChR2$^-$ and TH$^-$/ChR2$^+$ cells in all immunopositive cells, calculated from counts in all four animals. (**D**) An image of ChR2$^+$ projection fibers (green) in the dStr, showing dense projections of TH$^+$/VGLUT2$^+$ fibers in the ldStr, and very sparse projections in the mdStr. Dense fibers on the bottom left are projections to the NAc, where the most prominent glutamate cotransmission is observed. White squares outline regions shown at higher magnification on the right.

DOI: https://doi.org/10.7554/eLife.39786.019

VAChT showed high enrichment in ChIs (*Figure 10A*). D5R, TrpC3 and TrpC7 were enriched in ChIs, while mGluR1, mGluR5 and D1R were reduced (*Figure 10A*). This suggests that increased expression of TrpC3/7 and D5R, but not mGluR1, determines the discrete observation of slow EPSCs in ChIs.

When we compared gene expression relative to GAPDH between the mdStr and ldStr, D1, mGluR5 and TrpC3 expression showed significant differences between the mdStr and ldStr (*Figure 10B*). Subregional differences in gene expression were not due to differences in the numbers of ChIs, as their density was the same in the mdStr and ldStr (*Figure 10—figure supplement 1*). Considering the low expression of D5R and TrpC7 (*Figure 10B*), reduced expression of D1R in ChIs (*Figure 10A*) and the minor contribution of D1-like receptors and TrpC7 to the slow EPSC (*Figure 4*), high expression and enrichment of TrpC3 in ChIs appears to be crucial for the slow EPSC. However, the subregional difference in TrpC3 expression between mdStr and ldStr was significantly smaller

than the difference in the size of the slow EPSC (viz. *Figure 3*), so differential TrpC3 expression was not the determinant of the prominence of the slow EPSC in the ldStr. Thus, the principal determinant of the medial-lateral difference appears to be the projections of VGLUT2⁺ DA neurons, while the discrete observation of the slow EPSCs in ChIs appears to be due to postsynaptic TrpC3 expression.

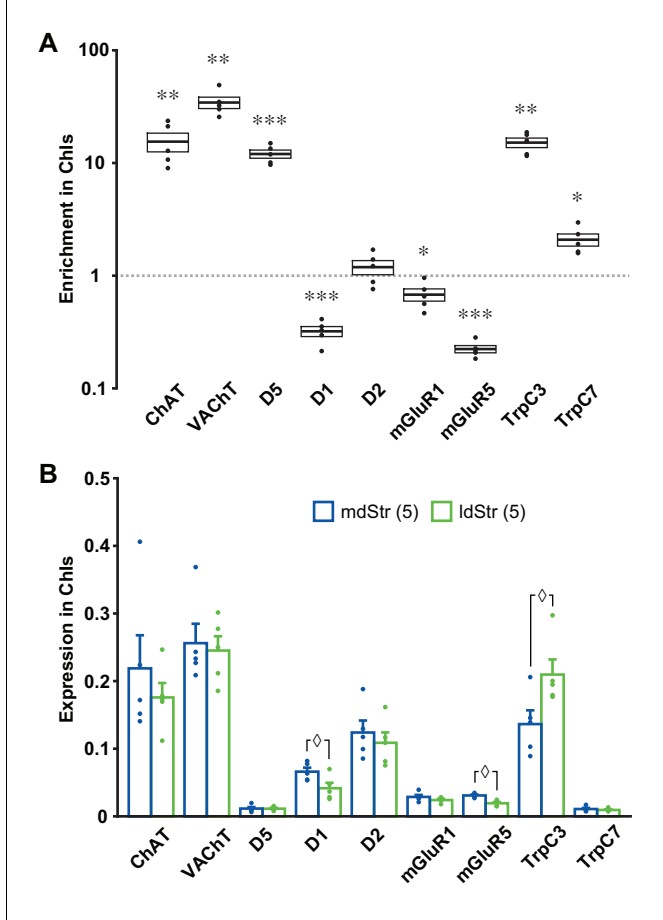

**Figure 10.** Expression of slow EPSC mediators in ChIs. qPCR measurements of ChAT, VAChT, DA receptors, mGluR1/5 and TrpC 3/7 expression in dStr ChIs. (A) Enrichment in ChIs is shown relative to whole Str RNA (input sample). A ratio of 1 indicates that expression is the same in ChIs and all dStr cells. In box plot, the middle lines and horizontal outlines of boxes indicate means and SEM, respectively. Statistical significance was examined with one-sample t-test to 1 (no enrichment). See *Figure 10—source data 1* for exact p values for each gene. *, ** and *** indicate p<0.05, p<0.01 and p<0.001, respectively, for differences from a ratio of 1. n = 5 replicates (three animals per replicate). (B) Subregional differences in expression in ChIs in the mdStr (blue) and ldStr (green) are shown, relative to a housekeeping gene (GAPDH). Regional differences were examined using a general linear model multivariate analysis. See *Figure 10—source data 1* for exact F values and p values. ◊ indicates p<0.05 for comparison between the mdStr and ldStr. n = 5 replicates (shown in parentheses). Dots show the result for each replicate. See also *Figure 10—figure supplement 1*.

DOI: https://doi.org/10.7554/eLife.39786.020

The following source data and figure supplement are available for figure 10:

**Source data 1.** (statistics for *Figure 10*).

DOI: https://doi.org/10.7554/eLife.39786.022

**Figure supplement 1.** Estimation of ChI density in the mdStr and ldStr.

DOI: https://doi.org/10.7554/eLife.39786.021

## Discussion

DA neurons differentially control ChIs in the dStr. They inhibit mdStr ChIs, while they inhibit and excite ldStr ChIs. The inhibition is D2 mediated, and shorter in ldStr ChIs due to an ensuing slow EPSC. The slow EPSC is mediated monosynaptically by glutamate cotransmission activating mGluR1 receptors, coupled to activation of TrpC3/7, along with DA activation of D1-like (D1/5) receptors. mGluR1 responses are observed only in ChIs in the ldStr, while SPNs show iGluR responses. Topographic projections of SN DA neurons to the ldStr mediate the glutamate responses in ChIs. Slow DA neuron driven EPSCs are seen discreetly in ldStr ChIs, due presynaptically to SN DA neuron glutamate cotransmission and postsynaptically to enriched expression of TrpC3, that together determine the unique responses of ldStr ChIs to DA neuron activity (*Figure 11*).

### Slow EPSCs in lateral dorsal striatum cholinergic interneurons

Optogenetic activation of DA neurons projecting to the dStr evokes a delayed excitation, of unclear mediation (*Straub et al., 2014*), which corresponds in magnitude and timing to the slow EPSC. Pharmacological isolation of the slow EPSC together with use of selective and more potent antagonists of group I mGluRs revealed mGluR1 and D1 as mediators of the slow EPSC. We used the highly potent and selective mGluR1 antagonist JNJ, at the maximum mGluR-selective concentration (*Fukunaga et al., 2007*). Less specific mGluR antagonists require higher concentrations that may mask mGluR1 effects through opposing actions at other mGluRs. While the JNJ and MTEP cocktail showed significant mGluR1 effects after 10 min, the D1 antagonist SCH23390 required 20 min to show significant action. SCH loses D1 selectivity at higher concentrations due to Ca2$^+$ channel blockade (*Guenther et al., 1994*), so longer application times are required.

The slow EPSC was not completely blocked by mGluR1 and D1 antagonists; the small residual EPSC could be due to incomplete antagonist action, or could be a minor still unspecified response mediated by a G-protein and TrpC3/7, as it was blocked by GDPβS or FFA. FFA has multiple actions besides TrpC3/7 antagonism; it is also a TrpC6 agonist and affects other ion channels. The lack of significant effects on fast glutamate EPSCs excluded effects on transmitter release through other ion channels, and complete blockade of slow EPSCs excluded a TrpC6 contribution to the slow EPSCs.

Optogenetic activation of DA neurons projecting to olfactory tubercle (OT) evokes a delayed D1-mediated excitation (*Wieland et al., 2014*). While the delayed excitation in the OT is solely D1 mediated, D1-like receptors mediate only part of the response in dStr ChIs. Since DA neuron glutamatergic projections to the mdStr are sparse, the small slow EPSCs seen in some mdStr ChIs are likely D1R mediated, pointing to slow D1-mediated excitation of ChIs across the Str. Both D1 and D5 receptors are expressed in ChIs, but DA application produces the same depolarization in wild type and D1R KO mice (*Centonze et al., 2003*), arguing that the D1 component of slow EPSCs is mainly D5R mediated. Although our observation of enriched D5R expression and reduced D1R expression in ChIs agree with previous observations (*Bergson et al., 1995*; *Lim et al., 2014*; *Yan et al., 1997*), we found a higher absolute D1R expression in ChIs than previously reported (*Bergson et al., 1995*; *Yan et al., 1997*); this could be due to methodological differences, species (rat vs. mouse) or animal age. Immunostaining would not necessarily reveal D1Rs in ChIs given the high D1R expression in dSPNs, which greatly outnumber ChIs (*Bergson et al., 1995*).

The slow EPSC in dStr ChIs is mainly mGluR1 mediated. Slow mGluR1 EPSCs were originally described in cerebellar Purkinje cells (*Batchelor and Garthwaite, 1997*) and ventral midbrain DA neurons (*Fiorillo and Williams, 1998*; *Shen and Johnson, 1997*); however, these slow EPSCs are seen only with spike trains, suggesting that they are mediated by glutamate spillover acting at extrasynaptic receptors. In contrast, slow EPSCs in ldStr ChIs are reliably evoked with single spikes, at low frequency, suggesting that mGluR1 receptors are closer to synaptic release sites in ChIs than in Purkinje cells or DA neurons. ChIs, as well as SPNs, express both mGluR1 and iGluRs, and iGluR responses are evoked in SPNs by stimulation of other glutamate inputs (*Ding et al., 2010*; *Johnson et al., 2017*), but not mGluR1 responses. Furthermore, mGluR1 is not enriched in ChIs. Thus, the different glutamate responses in ChIs and SPNs — mGluR1 in ChIs and iGluRs in SPNs — appear to depend on the differential distribution of the receptors at postsynaptic sites of DA neuron glutamate cotransmission in the two cell types.

Perfusion of group I mGluR agonists evokes a TrpC3/7 mediated depolarization in dStr ChIs in rat brain slice (*Berg et al., 2007*); our expression studies show that dStr ChIs express both mGluR1 and

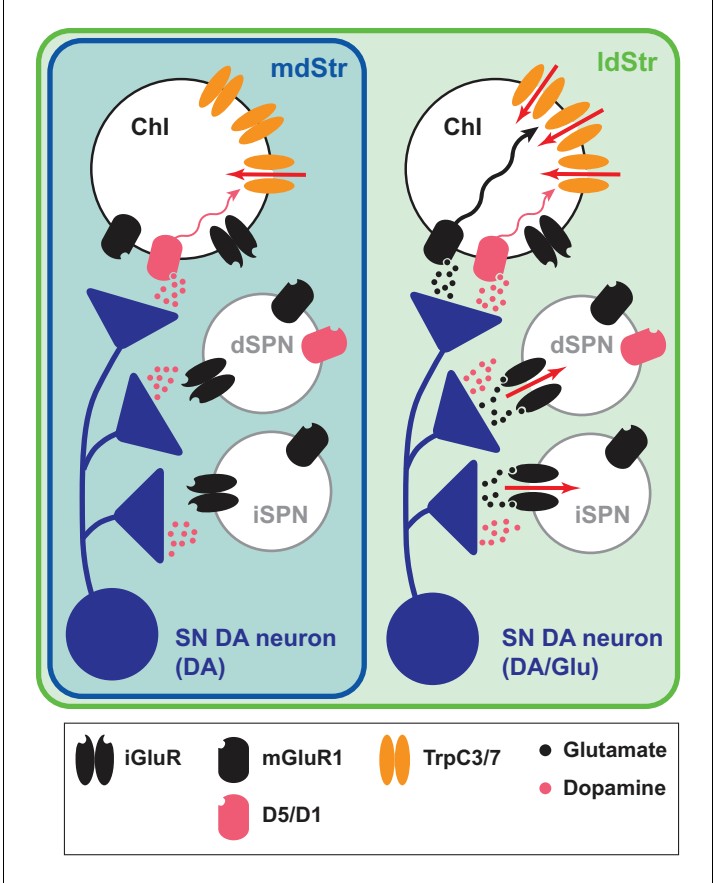

**Figure 11.** Summary of DA neuron excitatory synaptic responses in the dStr. Transmitter release sites are shown for one presynaptic terminal per postsynaptic Str cell type. Receptors distant from postsynaptic sites have little involvement in the synaptic transmission; modulatory effects of DA and glutamate are not shown. DA neurons evoke DA signals at all their synaptic connections in both the mdStr (outlined in blue) and ldStr (outlined in green), while they evoke glutamate signals only in the ldStr. mGluR1 activates cell signaling pathways in ldStr ChIs (wavy arrows) that trigger ion flux through TrpC channels (red arrows). The present results do not distinguish between DA and Glu release from the same or different vesicles or boutons.
DOI: https://doi.org/10.7554/eLife.39786.023

TrpC3. TrpC3 is also a major mediator of mGluR1-mediated slow EPSCs in cerebellar Purkinje cells (*Hartmann et al., 2008*), suggesting that ldStr ChIs share with Purkinje cells the same mGluR EPSC mechanism. mGluR1 is also expressed in ventral midbrain DA neurons and their terminals (*Fiorillo and Williams, 1998*; *Zhang and Sulzer, 2004*), and presynaptic mGluR1s reduce DA release in the Str (*Zhang and Sulzer, 2004*). Since activation of presynaptic receptors requires transmitter spillover (*Scanziani et al., 1997*), activation of presynaptic mGluR1 on DA neuron terminals by single spikes is less likely. However, when DA neurons fire in bursts, mGluR1-mediated presynaptic inhibition would limit glutamate cotransmission in the ldStr, and likely reduce temporal summation.

## Glutamate cotransmission from substantia nigra dopamine neurons

While DA neuron cotransmission consistently evokes iGluR EPSCs in the NAc medial shell, reports of EPSCs in the dStr have varied (*Chuhma et al., 2014*; *Mingote et al., 2015*; *Stuber et al., 2010*; *Tritsch et al., 2012*). Our present results reveal that iGluR EPSCs vary based on recording location, with small or no responses in the mdStr and larger responses in the ldStr. mGluR1 responses in ChIs are prominent in the ldStr. Thus, the ldStr — in addition to the NAc medial shell — is a hotspot of DA neuron glutamate cotransmission. Given the topography of DA neuron projections (*Haber et al., 2000*; *Ikemoto, 2007*), the ldStr should receive input solely from the SNc. Although our CTB results

showed the existence of non-topographical projections from VTA DA neurons to the ldStr, as has also been observed in rats (*Maurin et al., 1999*; *Pennartz et al., 2009*), these non-topographical VTA projections do not mediate glutamate cotransmission in the ldStr. Considering the very sparse expression of ZsGreen in the VTA following ldStr injections, the non-topographic projection appears to arise from non-glutamate cotransmitting DA neurons, which may contribute to D1- or D2-like receptor responses in ldStr ChIs. Our retrograde viral tracing showed that most glutamate cotransmission in the ldStr ChIs arises from SN DA neurons, where VGLUT2 expression has not been reported in adult mice (*Morales and Root, 2014*). This discrepancy could be due to the difficulty of detecting a small population of neurons with relatively low expression levels by in situ hybridization.

The INTRSECT strategy revealed that about a fifth of SN DA neurons coexpress VGLUT2, which was more than found by in situ hybridization (*Morales and Root, 2014*) or CAV2-FLEX-ZsGreen retrograde labeling. Similarly, the INTRSECT strategy revealed a few fibers in the mdStr, but there was no retrograde labeling with ZsGreen. This could be due to a lack of CAV receptors on mdStr projecting DA neurons, but was more likely due to the sparse distribution of mdStr projecting DA neurons and the more restricted volume of the CAV injection. While DA neurons capable of glutamate cotransmission constitute a minority of SN neurons, they elicit substantial excitation due to their relatively restricted projection area. The INTRSECT strategy showed TH/VGLUT2 colocalization in the VTA as well. Since TH$^+$/VGLUT2$^+$ neurons in the VTA made little contribution to the non-topographic projection to the ldStr, these coexpressing VTA neurons presumably project topographically to the NAc. There were ChR2$^+$/TH$^-$ neurons in the SN, apparently showing lower specificity of the INTRSECT strategy. This could be due to a combination of slightly higher non-specific expression of TH-driven gene expression than DAT-driven gene expression (*Lammel et al., 2015*), recombination efficacy/specificity of the particular INTRSECT virus, and limitations in identifying colocalization of cytosolic TH and membrane-targeted ChR2 (*Stuber et al., 2015*). Some non-specific expression, notwithstanding, the INTRSECT results showed substantial SN DA neuron glutamate projections to the ldStr, and sparse projections to the mdStr, consistent with the CAV2 results, showing that DA/ glutamate neurons in the SN project to the ldStr.

Our retrograde tracing studies revealed that about 40% of mdStr or ldStr projecting neurons are non-DAergic, presumably GABA-only or glutamate-only neurons (*Morales and Root, 2014*; *Morello and Partanen, 2015*). Glutamate-only neurons are seen in both the VTA and SNc (*Hnasko et al., 2012*; *Morales and Root, 2014*), and indeed we observed 30–40% TH$^-$/VGLUT2$^+$ neurons in both the VTA and the SN (SNc/SNr). Both TH$^+$/VGLUT2$^+$ and TH$^-$/VGLUT2$^+$ (glutamate-only) neurons projecting to the ldStr reside in the SN, and do not project to the mdStr, consistent with the recognized segregation of DA neurons projecting to the medial and lateral dStr (*Lerner et al., 2015*). Thus, glutamate cotransmitting DA neurons and glutamate-only neurons appear to project similarly to the dStr.

## Functional implications

DA neurons may engage ChIs as a hub to control Str circuits. Although ChIs comprise only 1–2% of striatal neurons, they exert a strong control of striatal circuits with their large axonal fields (*Kawaguchi et al., 1995*; *Kreitzer, 2009*). In response to reward-related cues or outcomes, dStr ChIs frequently show a burst-pause-burst firing pattern, coincident with DA neuron burst firing (*Morris et al., 2004*; *Schulz and Reynolds, 2013*). While the early burst is mediated by thalamic inputs (*Ding et al., 2010*), the pause appears to be principally D2R mediated. The subsequent rebound in firing is highly variable, suggesting that it is generated and modulated independently of the D2-mediated pause (*Schulz and Reynolds, 2013*). Multiple mechanisms likely contribute to the rebound firing, one of which is the DA neuron mediated slow EPSC. The presence of the slow EPSC in the ldStr, and not the mdStr, likely contributes to subregional variability in ChI rebound firing.

The slow EPSC is likely to shorten or cancel the D2R mediated firing pause in ldStr ChIs. Although the circuit functions of ChI pauses have not been totally elucidated (*Zhang and Cragg, 2017*), there are several suggestive physiological observations. Silencing ChIs reduces spontaneous IPSCs in SPNs, while ChI firing increases sIPSCs (*de Rover et al., 2002*), presumably through activation of a class of GABA interneurons (*English et al., 2011*). Cortical or thalamic glutamatergic inputs are reduced by activation of presynaptic muscarinic ACh receptors (mAChR) (*Ding et al., 2010*; *Higley et al., 2009*), and a single ChI spike is sufficient to reduce cortical glutamate EPSCs in SPNs (*Pakhotin and Bracci, 2007*) 391-400). Thus, we can presume that pausing ChIs provides a window

facilitating excitation of SPNs by lessening GABA inhibition, and simultaneously accentuating glutamate inputs by removing mAChR-mediated inhibition. The delayed increase of firing in ChIs caused by the slow EPSCs may regulate the excitation window for SPNs in the ldStr.

At a local circuit level, cholinergic tone in the dStr modifies frequency dependence of DA release (*Threlfell and Cragg, 2011*), excitatory input strength, including long-term plasticity (*Tanimura et al., 2016*), and GABA tone by affecting Str interneurons and SPNs (*English et al., 2011*). DA neuron tonic activity may affect basal cholinergic tone differentially in the mdStr and ldStr due to the mGluR component in the ldStr. Str ChIs play important roles in behavioral flexibility (*Aoki et al., 2015*; *Okada et al., 2014*); chemogenetic activation of ldStr ChIs enhances switching habits (*Aoki et al., 2018*). Therefore, the slow EPSC may modulate habit switching by transiently countering tonic DA neuron DA modulation of ldStr ChIs.

While DA neurons elicit excitatory signals via D1R and inhibitory signals via D2R across the dStr, subregional differences in the dStr involve differences in the strength of the signals. In contrast, glutamate cotransmission varies in timing and synaptic mechanism, with hotspots of fast cotransmission in the NAc medial shell and ldStr and slow excitation limited to ChIs in the ldStr. Thus, DA neuron glutamate cotransmission appears to be a major driver of subregional heterogeneity in DA neuron actions across the Str.

# Materials and methods

## Key resources table

| Reagent type (species) or resource | Designation | Source or reference | Identifiers | Additional information |
|---|---|---|---|---|
| Genetic reagent (*M. musculus*) | B6.SJL-*Slc6a3*$^{tm1.1(cre)Bkmn}$/J | Jackson Laboratories | RRID:IMSR JAX:006660 | henceforth DAT$^{IREScre}$ |
| Genetic reagent (*M. musculus*) | B6.Cg-Gt(ROSA)26 Sor$^{tm32(CAGCOP4*H134R/EYFP)Hze}$/J | Jackson Laboratories | RRID:IMSR JAX:024109 | henceforth Ai32 |
| Genetic reagent (*M. musculus*) | B6.129 × 1-Gt(ROSA)26Sor$^{tm1(EYFP)Cos}$/J | Jackson Laboratories | RRID:IMSR JAX:006148 | henceforth R26-stop-EYFP |
| Genetic reagent (*M. musculus*) | B6.Cg-Tg(RP23-268L19-EGFP)2Mik/J | Jackson Laboratories | RRID:IMSR JAX:007902 | henceforth ChAT-eGFP |
| Genetic reagent (*M. musculus*) | B6.Cg-Tg(Drd1a-td Tomato)6Calak/J | Jackson Laboratories | RRID:IMSR JAX:016204 | henceforth D1-tdTomato |
| Genetic reagent (*M. musculus*) | Tg(Drd2-EGFP) S118Gsat/Mmnc | GENSAT | RRID:MMRRC _000230-UNC | henceforth D2-EGFP |
| Genetic reagent (*M. musculus*) | B6.129S4-Slc17a6$^{tm1Rpa}$/J | Jackson Laboratories | RRID:IMSR JAX:007583 | henceforth VGLUT2$^{lox}$ |
| Genetic reagent (*M. musculus*) | Slc17a6$^{tm2(cre)Lowl}$/J | Jackson Laboratories | RRID:IMSR JAX:016963 | henceforth VGLUT2$^{IREScre}$ |
| Genetic reagent (*M. musculus*) | TH$^{2A-flpo}$ | PMID:30104732 | | |
| Genetic reagent (*M. musculus*) | B6.FVB(Cg)-Tg(Chat-cre)GM60Gsat/Mmucd | GENSAT | RRID:MMRRC _030869-UCD | henceforth ChAT$^{cre}$ |
| Genetic reagent (*M. musculus*) | B6N.129-Rpl22$^{tm1.1Psam}$/J | Jackson Laboratories | RRID:IMSR JAX:011029 | henceforth RiboTag |

*Continued on next page*

*Continued*

| Reagent type (species) or resource | Designation | Source or reference | Identifiers | Additional information |
|---|---|---|---|---|
| Antibody | anti-TH (mouse monoclonal) | Millipore | RRID:AB_2201528; Cat.#:MAB318 | IHC (1:5000-1:10000) |
| Antibody | anti-CTB (goat polyclonal) | List Biological Laboratories | RRID:AB_10013220; Cat.#:703 | IHC (1:1000) |
| Antibody | anti-EGFP (rabbit polyclonal) | Millipore | RRID:AB_91337; Cat.#:AB3080 | IHC (1:2000) |
| Antibody | anti-ChAT (goat polyclonal) | Millipore | RRID:AB_2079751; Cat.#:AB144P | IHC (1:1000) |
| Antibody | anti-HA.11 Epitope Tag | Biolegend | RRID:AB_2565334; Cat.#:901515 | IP (1:160) |
| Antibody | anti-mouse Alexa Fluor 647 | ThermoFisher | RRID:AB_2535813; Cat.#:A-21245 | IHC, secondary (1:200) |
| Antibody | anti-goat Alexa Fluor 488 | ThermoFisher | RRID:AB_2534102; Cat.#:A-11055 | IHC, secondary (1:200) |
| Antibody | anti-goat Alexa Fluor 555 | ThermoFisher | RRID:AB_2535853; Cat.#:A-21432 | IHC, secondary (1:200) |
| Antibody | anti-rabbit Alexa Fluor 488 | ThermoFisher | RRID:AB_2535792; Cat.#:A-21206 | IHC, secondary (1:200) |
| Recombinant DNA reagent | AAV5-EF1a-DIO-hChR2(H134R)-EYFP | Vector Core Facility, University of North Carolina | | www.med.unc.edu/ genetherapy/vectorcore /in-stock-aav-vectors/deisseroth |
| Recombinant DNA reagent | AAV-hSyn-Con/Fon hChR2(H134R)-EYFP-WPRE | Vector Core Facility, University of North Carolina | | www.med.unc.edu/genetherapy /vectorcore/in-stock-aav-vectors/deisseroth |
| Recombinant DNA reagent | CAV2-FLEX-ZsGreen | Larry Zweifel (University of Washington) | | depts.washington.edu/ zweifel/resources.html |
| Peptide, recombinant protein | TTX (tetrodotoxin) | Ascent Scientific | Cat.#:Asc-055 | |
| Peptide, recombinant protein | CTB (cholera toxin subunit B) | List Biological Laboratories | Cat.#:104 | |
| Peptide, recombinant protein | Dynabeads Protein G | ThermoFisher | Cat.#:10004D | |
| Commercial assay or kit | RNeasy Micro Kit | Qiagen | Cat.#:74004 | |
| Commercial assay or kit | RT2 First Strand Kit | Qiagen | Cat.#:330404 | |
| Commercial assay or kit | RT2 SYBR Green qPCR Mastermix | Qiagen | Cat.#:330504 | |
| Commercial assay or kit | Custom RT2 Profiler PCR Array, 96 well | Qiagen | Cat.#:330171 (CLAM23840) | |
| Commercial assay or kit | Quanti-iT RiboGreen RNA Assay Kit | ThermoFisher | Cat.#:R11490 | |
| Chemical compound, drug | S-(-)-Sulpiride | Tocris | Cat.#:0895 | henceforth Sulpiride |
| Chemical compound, drug | CNQX | HelloBio; Tocris | Cat.#:HB0204; Cat.#:0190 | |
| Chemical compound, drug | APV | HelloBio; Tocris | Cat.#:HB0225; Cat.#:0106 | henceforth D-AP5 |
| Chemical compound, drug | SR95531 | HelloBio; Tocris | Cat.#:HB0901; Cat.#:1262 | henceforth gabazine |

*Continued on next page*

*Continued*

| Reagent type (species) or resource | Designation | Source or reference | Identifiers | Additional information |
|---|---|---|---|---|
| Chemical compound, drug | 4-aminopyridine | SigmaAldrich | Cat.#:A0152 | henceforth 4-AP |
| Chemical compound, drug | GDPβS | SigmaAldrich | Cat.#:G7637 | |
| Chemical compound, drug | SCH23390 | HelloBio; Tocris | Cat.#:HB1643; Cat.#:0925 | henceforth SCH |
| Chemical compound, drug | JNJ16259685 | Tocris | Cat.#:2333 | henceforth JNJ |
| Chemical compound, drug | MTEP | Tocris | Cat.#:2921 | |
| Chemical compound, drug | Pyr3 | Tocris | Cat.#:3751 | |
| Chemical compound, drug | Flufenamic acid | Tocris | Cat.#:4522 | henceforth FFA |
| Chemical compound, drug | CGP54626 | Tocris | Cat.#:1088 | |
| Chemical compound, drug | Mecamylamine | Tocris | Cat.#:2843 | |
| Chemical compound, drug | Scopolamine | Tocris | Cat.#:1414 | |
| Chemical compound, drug | QX-314 | SigmaAldrich | Cat.#:L5783 | |
| Software, algorithm | pClamp 10 | Molecular Devices | RRID:SCR_011323 | |
| Software, algorithm | Axograph X | Axograph Sciences | RRID:SCR_014284 | |
| Software, algorithm | SPSS 23 | IBM | RRID:SCR_002865 | |
| Software, algorithm | JASP 0.8.6 | JASP team, 2018 | RRID:SCR_015823 | jasp-stats.org |
| Software, algorithm | G*Power 3.1 | Heinrich Heine University Düsseldorf | RRID:SCR_013726 | |
| Software, algorithm | Igor Pro 6 | WaveMetrics | RRID:SCR_000325 | |
| Software, algorithm | MATLAB R2014b | MathWorks | RRID:SCR_001622 | |

## Mice

Mice were handled in accordance with the guidelines of the National Institutes of Health *Guide for the Care and Use of Laboratory Animals*, under protocols approved by the Institutional Animal Care and Use Committee of New York State Psychiatric Institute. Mice were group housed and maintained on a 12 hr light/dark cycle. All slice/tissue preparations were done during the light phase. Food and water were supplied *ad libitum*. Postnatal day (P) 57–111 mice, male and female, were used.

DAT (*Slc6a3*)-internal ribosome entry site (IRES) cre (DAT[IREScre]) mice (*Bäckman et al., 2006*) (Jackson Laboratories; RRID:IMSR_JAX:006660) were mated with ROSA26-floxSTOP-CAG-ChR2-EYFP (Ai32) (Jackson Laboratories; RRID:IMSR_JAX:024109) to achieve selective stimulation of DA neuron terminals in the Str. To achieve stable stimulation, ChR2-EYFP homozygous mice were used. DAT[IREScre];R26-stop-EYFP (Jackson Laboratories; RRID:IMSR_JAX:006148) double mutant mice were used as controls. For identification of dSPN, iSPN and ChIs, mice with fluorescent genetic markers for each neuron type, D1-tdTomato (Jackson Laboratories; RRID:IMSR_JAX:016204), D2-EGFP (GENSAT; RRID:MMRRC_000230-UNC) or ChAT-eGFP (Jackson Laboratories; RRID:IMSR_JAX:

007902), respectively were bred with DAT[IREScre];ChR2-EYFP double mutant mice. For conditional knockout (cKO) of VGLUT2 (*Slc17a6*), floxVGLUT2 mice (*Hnasko et al., 2010*) (Jackson Laboratories; RRID:IMSR_JAX:007583) were bred with DAT[IREScre];ChR2-EYFP mice. Triple mutant of DAT[IREScre]; ChR2-EYFP;VGLUT2[lox/lox] mice (cKO) and control DAT[IREScre];ChR2-EYFP;VGLUT2[+/+] were used for experiments. For retrograde tracer injection, hemizygous VGLUT2[IREScre] mice (Jackson Laboratories; RRID:IMSR_JAX:016963) were used. For TH/VGLUT2 colocalization study, double hemizygous VGLU-T2[IREScre]; TH-2A-Flpo (TH[2A-flpo]) mice were used (*Poulin et al., 2018*). For ChI gene expression studies, RiboTag mice (Jackson Laboratories; RRID:IMSR_JAX:011029) were bred with transgenic ChAT[cre] mice (GENSAT; RRID:MMRRC_030869-UCD) to express hemagglutinin (HA) epitope tagged ribosomal protein L22 in ChIs.

Mice used for the experiments were on a C57BL6J background or a C57BL6J/129Sv mixed background, more than five times back crossed to C57BL6J and kept inbred. D2-EGFP mice, originally on a FVB background, were backcrossed to C57BL6J at least eight times. VGLUT2[IREScre] mice, originally on a mixed C57BL6J;FVB;129S6 background, were backcrossed 3–5 times to C57BL6J. RiboTag mice were on a C57BL6NJ background.

## Slice electrophysiology

Mice (P60-81) were anesthetized with a ketamine (90 mg/kg)/xylazine (7 mg/kg) mixture. After confirmation of full anesthesia, mice were decapitated and brains quickly removed in ice-cold high-glucose artificial cerebrospinal fluid (ACSF) (in mM: 75 NaCl, 2.5 KCl, 26 NaHCO$_3$, 1.25 NaH$_2$PO$_4$, 0.7 CaCl$_2$, 2 MgCl$_2$ and 100 glucose, pH 7.4) saturated with mixture of 95% O2% and 5% CO2 (carbogen). Coronal sections of the striatum were cut, 300 μm thick, with a vibrating microtome (VT1200S, Leica), incubated in high glucose ACSF at room temperature for at least 1 hr for recovery, then transferred to the recording chamber (submerged, 500 μl volume) on the stage of an upright microscope (BX61WI, Olympus), continuously perfused with standard ACSF (in mM: 125 NaCl, 2.5 KCl, 25 NaHCO$_3$, 1.25 NaH$_2$PO$_4$, 2 CaCl$_2$, 1 MgCl$_2$ and 25 glucose, pH 7.4) saturated with carbogen. ChR2-EYFP, D2-EGFP or ChAT-eGFP expression was confirmed by field illumination with a 470 nm LED; D1-tdTomato expression was confirmed with 530 nm LED illumination (DC4100, Thorlabs). Recorded neurons were visualized using enhanced visible light differential interference contrast (DIC) optics with a scientific c-MOS camera (ORCA-Flash4.0LT, Hamamatsu Photonics).

In DAT[IREScre]:Ai32 mice, ChIs were identified visually by large soma size, confirmed by spontaneous firing, shallow resting membrane potentials (around −60 mV) and voltage sag by −400 pA current injection (700 msec duration) (*Chuhma et al., 2014*). Recording patch pipettes were fabricated from standard-wall borosilicate glass capillary with filament (World Precision Instruments). Pipette resistance was 3–7 MΩ. Composition of intracellular solution for firing and the slow EPSC recording was (in mM): 135 K$^+$-methane sulfonate (MeSO$_4$), 5 KCl, 2 MgCl$_2$, 0.1 CaCl$_2$, 10 HEPES, 1 EGTA, 2 ATP and 0.1 GTP, pH 7.25. For G-protein coupled receptor blockade, GTP was replaced with 0.5–1 mM GDPβS. GDPβS pipette solution was made from powder just before recording, kept on ice and discarded after 2 hr. For fast glutamate EPSC recording, Cs$^+$-based pipette solution was used; K$^+$-MeSO$_4$ was replaced with Cs$^+$-MeSO$_4$, with QX314 (lidocaine N-ethyl bromide) 5 mM. Both voltage and current clamp recordings were performed with an Axopatch 200B amplifier (Molecular Devices). Holding potential was −70 mV. Current clamp recordings were done in fast current clamp mode. Series resistance (8–26 MΩ) was compensated online by 70–75%. Liquid junction potentials (10–12 mV) were adjusted online. Synaptic responses were evoked with five msec field illumination with a high-power blue LED (Thorlabs) delivered either as a single pulse at 0.1 Hz, or in a train of five pulses at 20 Hz, repeated at 30 s intervals. Since some slow EPSC recording showed a build up with repeated stimulation, the control pre-drug responses were recorded after the EPSC had reached a plateau.

For pharmacological studies, drugs were delivered by perfusion. For wash off of the effects of mGluR antagonists and D1-antagonist on isolated slow EPSCs, slices were perfused with regular ACSF for 15–25 min, followed by control antagonists (CNQX, D-APV, SR95531 and sulpiride) for at least 5 min. Recording from VGLUT2 cKO mice and control mice were done genotype blind. All the recordings were done at 32–34°C (TC 344B Temperature Controller, Warner Instruments). Data were filtered at 5 kHz using a 4-pole Bessel filter, digitized at 5 kHz (Digidata 1550A, Molecular Devices) and recorded using pClamp 10 (Molecular Devices; RRID:SCR_011323).

## Data analysis for electrophysiology

Electrophysiological data were analyzed with Axograph X (Axograph Science; RRID:SCR_014284). Firing z score during or post train stimulation was calculated as the difference of average firing frequency during (0–0.4 s from train onset) or post train (0.5–0.9 s from train onset), minus the average baseline firing frequency, divided by the standard deviation of baseline firing frequency. Plus score indicates increase of firing from the baseline, while minus score indicates decrease of firing. Since ChIs were firing continuously and it was hard to identify resting state of membrane, we regarded averaged membrane potentials after action potentials were truncated as the resting membrane potentials. PSC evaluation was done from averages made from 10 consecutive traces. Slow EPSCs were evaluated by measuring charge transfer in a 1 s or 1.5 s window ('area under the curve') for more reliable comparison. Since synaptic responses to the first stimulus after a long interval were artificially large, we discarded them. Data are shown as mean ± S.E.M.

## Fast-scan cyclic voltammetry in slice

Fast-scan cyclic voltammetry (FSCV) recordings were done in P72-93 DAT$^{IREScre}$;ChR2-EYFP mice. Slice preparation and recording environment were the same as described in the Slice electrophysiology section. DA release was evoked with either a single pulse (five msec duration) or a train (5 pulses at 20 Hz; 5 ms duration pulses) of field illumination of blue LED. Photostimulation trains was applied at intervals of 1–2 min. DA release was measured using carbon fiber electrodes placed 30–100 μm below the slice surface. Carbon fibers was calibrated in 1 μM DA after each experiment. A triangular voltage wave (−400 to + 800 mV at 300 V/sec vs. Ag/AgCl) was applied to the electrode at 10 Hz and the resulting currents were recorded using an Axopatch 200B (Molecular Devices), filtered at 10 kHz with a 4-pole Bessel filter. Offset currents were not applied to avoid saturation of the amplifier. DA currents were confirmed by examination of the cyclic voltammograms for the signature DA oxidation current waveform. Traces of DA release were obtained by plotting current measured at the potential of peak DA oxidation for each voltage command as a function of time, in Igor Pro (Wavemetrics; RRID:SCR_000325) using custom routines (Eugene Mosharov, sulzerlab.org). Data were analyzed with MATLAB R2014b (Mathworks; RRID:SCR_001622).

## Viral vector and retrograde tracer injection

For selective ChR2 expression in ventral midbrain DA neurons, we injected AAV genotype five encoding ChR2 fused to EYFP (AAV5-EF1a-DIO-ChR-EYFP; UNC Vector Core) into DAT$^{IREScre}$ mice. Mice (P33-36, n = 2 animals) were anesthetized with the ketamine/xylazine mixture. A glass pipette (PCR micropipettes, Drummond), pulled to a tip diameter ~ 20 μm, was lowered to just above the ventral midbrain (coordinates relative to bregma: −3.3 mm antero-posterior (AP), −4.3 mm dorso-ventral (DV), and 0.5 mm lateral (L) for the VTA and L 1.3 mm for the SN) and pressure injections of 0.5 μl of AAV5-EF1a-DIO-ChR2-EYFP (titer 1.5 × 10$^{12}$ virus molecules/ml) were made unilaterally in two locations. The pipette was left in place for ~ 3 min to minimize back flow along the injection tract, then withdrawn, and the mouse allowed to recover. Slice recordings were done 28–30 days post injection.

For retrograde tracing, 1 μl of cholera toxin B subunit (CTB) solution (10% dilution with ddH$_2$O; List Biological Laboratories), or 0.2 μl CAV2-FLEX-ZsGreen (titer 3 × 10$^{12}$; from Larry Zweifel, Univ Washington) was injected into medial (AP + 0.8 mm, L + 1.3 mm, DV −2.2 mm) or lateral (AP + 0.8 mm, L + 2.5 mm, DV −2.2 mm) dStr of VGLUT2$^{IREScre}$ mice (for CTB: P59-96, n = 5 animals; for CAV2: P65-81, n = 7 animals).

For visualization of glutamate cotransmitting DA neurons using the INTRSECT strategy, 1 μl of AAV-hSyn-Con/Fon-hChR2 (H134R)-EYFP-WPRE (UNC Vector Core; titer 2.3 × 10$^{12}$) was injected into the ventral midbrain of VGLUT2$^{IREScre}$; TH$^{2a-flpo}$ mice (P29, n = 4 animals), with the same SN coordinates as above.

## Immunohistochemistry

Wild type C57BL6J mice (P72-93) for ChI cell counts, mice 3 days after injection of CTB, 3 weeks after injection of CAV2, or 4 weeks after injection of INTRSECT virus were anesthetized with ketamine/xylazine and perfused with cold phosphate buffered saline (PBS), followed by 4% paraformaldehyde (PFA). Brains were removed and post-fixed for 2–16 hr in 4% PFA. Coronal sections, 50 μm

thick, were cut using a vibrating microtome (Leica VT1200S), and stored in a cryoprotectant solution (30% glycerol, 30% ethylene glycol in 0.1 M Tris HCl, pH 7.4) at −20°C until processing. Sections were washed in PBS and incubated in glycine (100 mM) for 30 min to quench aldehydes. Non-specific binding was blocked with 10% normal goat serum (NGS; Millipore) in 0.1 PBS Triton X-100 (PBS-T) for 2 hr. The sections were incubated with primary antibodies in 0.02% PBS-T and 2% NGS for 24 hr, at 4°C on a shaker. Primary antibodies were: anti-TH (1:5,000–10,000 dilution, mouse monoclonal, Millipore, RRID:AB_2201528), anti-CTB (1:1000 dilution, List Biological Laboratories, RRID:AB_10013220), anti-EGFP (1:2000 dilution, rabbit polyclonal, Millipore, RRID:AB_91337) and anti-choline acetyltransferase (ChAT; 1:1000 dilution, goat polyclonal, Millipore, RRID:AB_2079751). Sections were then washed with PBS and secondary antibodies were applied for 45 min in 0.02% PBS-T at room temperature. Secondary antibodies (1:200 dilution; ThermoFisher Scientific) were: anti-mouse Alexa Fluor 647 (RRID:AB_2535813), anti-goat Alexa Fluor 488 (RRID:AB_2534102), anti-goat Alexa Fluor 555 (RRID:AB_2535853) and anti-rabbit Alexa Fluor 488 (RRID:AB_2535792). Sections were mounted on gelatin subbed slides (Southern Biotech) and cover slipped with Prolong Gold aqueous medium (ThermoFisher Scientific) and stored at 4°C.

## Imaging and cell counts

For retrograde tracer injection, tiled images were obtained with an AxioImager.M2 fluorescence microscope and Zen software (Zeiss) using a 20x objective. Images were taken in 1 μm steps to subtend the entire thickness of slices (18–22 images per slice), and each z-section image was examined for immuno- or tracer-fluorescence. The SN and VTA were delineated based on TH staining and mouse brain atlas (Paxinos and Franklin, 2008). SN pars compacta (SNc) and pars reticulata (SNr) were not delineated, since the boundaries between SNc and SNr was not clear in caudal sections. While the retrorubral field (RRF) was delineated (see Supplemental Figure 3), cell counts in the region were not included. Cell counts from each midbrain section were summed per animal and then averaged across animals according to injection location. For CTB injected animals, CTB$^+$ neurons were counted in every fifth section (sampling interval 250 μm). For CAV2 injected animals, all ZsGreen$^+$ cells in all sections containing VTA/SN DA neurons were counted, since the number of labelled cells was small. The number of TH$^+$ retrogradely labelled cells (TH$^+$/ZsGreen$^+$ or TH$^+$/CTB$^+$) and TH$^-$ cells (TH$^-$/ZsGreen$^+$ or TH$^-$/CTB$^+$) were counted, and percent colocalization calculated for all tracer+ neurons. For the INTRSECT study, neurons were counted in every second section (sampling interval 100 μm) in the SN, counting SNc and SNr together.

For estimates of the density of ChIs in the dStr, the dStr was split into medial and lateral parts based on previous reports (Lerner et al., 2015; Voorn et al., 2004), and ChAT$^+$ cells were counted using the Optical Fractionator Probe in Stereo Investigator (MBF Bioscience) using a 10x objective. In each brain, 17–18 slices were analyzed. Stereo Investigator's Cavalieri Estimator Probe with a 100 × 100 μm grid was used to determine the volume of the subregions. Stereological studies were performed unilaterally.

## Extraction of ChI mRNA and qPCR

We used RiboTag immunoprecipitation (IP) (Sanz et al., 2009), as modified by Lesiak et al. (Lesiak et al., 2015). Double hemizygous RiboTag;ChAT$^{cre}$ mice (P57-76) were anesthetized with ketamine/xylazine. After decapitation, brains were quickly removed in ice-cold PBS. Thick coronal sections of the dStr were cut with a razor blade, and divided into mdStr and ldStr segments; to avoid contamination from cholinergic neurons in the septum or pallidum, only the ldStr was sampled in the caudal most section. Tissue from three mice was gathered to make one replicate in order to obtain sufficient mRNA from ChIs. Tissue was homogenized at 5% w/v in homogenization buffer (HB: Tris pH 7.4 50 mM, KCl 100 mM, MgCl$_2$12 mM and NP-40 1%) supplemented with protease inhibitors (SigmaAldrich), RNase inhibitor (200 U/ml, Promega), DTT (1 mM, SigmaAldrich) and cycloheximide (100 μg/ml, SigmaAldrich), and then centrifuged at 10,000 x g for 10 min at 4°C. Supernatant, 12.5 μl for each segment, was set aside as the input fraction (control sample for all Str cells) and stored at −80°C. The remaining supernatant was diluted to 50% with HB and incubated with anti-HA.11 epitope tag antibody (1:160 dilution, Biolegend) on a tube rotator for 4 hr at 4°C. Then Dynabeads Protein G (15 mg/ml; ThermoFisher Scientific) was added to the supernatant and incubated on the tube rotator overnight at 4°C. The Dynabead suspension was put on a magnet rack (Promega) to isolate

the beads, which were then washed three times with high-salt buffer (Tris pH 7.4 50 mM, KCl 300 mM, MgCl2 12 mM, NP-40 1%, DTT 1 mM, cycloheximide 100 µg/l). After the final wash, each sample of beads was resuspended in 350 µl RLT buffer (RNeasy Micro Kit, Qiagen) with β-mercaptoethanol (bME; 10 µl/ml, Gibco). The suspension was then vortexed at full speed for 30 s, and put on the magnetic rack again to remove the beads, and the supernatant was then used as the immunoprecipitation (IP) fraction. Similarly, 350 µl RLT buffer with bME was added to the input fraction, which was vortexed for 30 s and the RNA extracted. Both IP and input samples were eluted in 17 µl water.

After extraction, RNA was quantified using the Quant-iT RiboGreen RNA Assay Kit (ThermoFisher Scientific). The measured amount of RNA, in a volume of 17 µl, was in the range of 1.7–22.4 ng for IP samples, and 104–609 ng for input samples. RNA was reverse transcribed, from 16 µl of the 17 µL RNA solution, with the RT2 First Strand Kit (Qiagen). The resulting cDNA was stored at −20°C pending quantitative PCR (qPCR) determinations. qPCR was performed in Custom RT2 Profiler PCR Arrays (Qiagen, 96 well, #330171, CLAM23840) using RT2 SYBR Green qPCR Mastermix (Qiagen). In addition to the genes of interest, mGluR1, mGluR5, TrpC3 and TrpC7, other genes analyzed included ChAT and VAChT as IP controls, and D1, D2 and D5 receptors as genes of known differential expression in ChIs. GAPDH and β-actin were measured as housekeeping genes. RT controls included a positive PCR control and negative genomic DNA control. cDNA from IP samples was used for PCR without dilution, while cDNA from input samples was diluted 1:1 (with water). PCR was done with a CFX96 Touch thermocycler (BioRad), following a cycle protocol of 95°C for 10 min, 40 cycles of 95°C for 15 s and 60°C for 1 min, followed by a melting curve. Genomic DNA controls were not amplified in any samples. Expression was normalized to GAPDH using Ct values ($\Delta$Ct). Differences in expression between dStr regions were expressed as $2^{-\Delta Ct}$. For analysis of enrichment of expression in ChIs, the $\Delta$Ct's of the IP sample and the input sample were compared using $\Delta\Delta$Ct, and the fold-change $2^{-\Delta\Delta Ct}$ calculated. A fold-change greater than one reflects enrichment in the IP sample. One replicate was omitted as it had more than a 3 SD deviation in $2^{-\Delta\Delta Ct}$.

## Statistical analysis

Sample size estimation was done with G*Power 3.1 (Heinrich Heine University, Dusseldorf; RRID: SCR_013726), setting $\alpha$=0.05, and power = 0.9. Effect size was estimated from previous experiments. For t-tests, effect size for experiments anticipating complete blockade (e.g. CNQX effects on fast EPSCs) was set at 2.5. For experiments anticipating partial blockade or enhancement (*e.g.* sulpiride effects on PSCs), effect size was set at 1.5. These analyses required an n per group of 5 or 7, respectively. For repeated measures ANOVA (including mixed ANOVA), effect sizes for complete blockade and partial blockade were set at 0.8 and 0.5, giving a required n per group of 4 or 7, respectively. For regional comparisons (independent sample comparisons), we set an effect size of 0.8, based on previous recordings from different striatal subregions, giving a required n per region of 10. We did not use non-parametric tests, because (1) generally non-parametric tests are less sensitive with small sample numbers (e.g. less than 20), (2) the variables we measured were continuous numeric variables (not ranked variables), which are likely to show a normal distribution, and (3) non-parametric alternatives do not exist for some parametric tests (*e.g.* mixed ANOVA).

Statistical analysis was done with SPSS 23 (IBM; RRID:SCR_002865) and JASP ver 0.8.6 (JASP Team, 2018; jasp-stats.org; RRID:SCR_015823). Comparisons of two values were done using a t-test. When sample size (biological replicate number) was smaller than 10 or variances were not equal, a t-test without assumption of equal variances (Welch's t-test) was used. For evaluation of drug effects, comparison was made on a percent basis to the pre-drug response (100%) using a one-sample t-test. For more than three variable comparisons, ANOVA was used. In repeated-measures ANOVA, when sphericity was violated, Greenhouse-Geisser correction was conducted. For gene expression studies, regional differences were compared using a general linear model multivariate analysis. For one-way or two-way ANOVA, Scheffe's post-hoc test was used to identify significant differences. For mixed ANOVA, when significant interactions were found, post-hoc t-tests were done for between-subject effects. For CTB cell counts, $\chi^2$ test was used. For CAV2 counts, a mixed ANOVA was used, as $\chi^2$ failed with zeros in some cells. p values smaller than 0.05 were regarded as significant. Data are reported as mean ± S.E.M., unless otherwise noted. In the graphs, dots show the average measurements for each biological replicate, and bars show the mean and S.E.M. for all biological replicates. Exact values of n, what n represents, p values, and F values for ANOVA tests are presented in the figure legends. Numbers of animals used for electrophysiological recordings are indicated in the

figure legends. For the main electrophysiological experiments, no more than 3 cells were recorded per animal; for the VGLUT2 cKO experiments and AAV-DIO-ChR2 injection experiments, 3 to 6 cells were recorded per animal. p values are shown to the third decimal place, so p=0.000 reflects p<0.001.

## Acknowledgements

We thank Larry Zweifel for CAV2-FLEX-ZsGreen, and Timothy Cheung and Vlad Velicu for technical help and advice.

## Additional information

### Funding

| Funder | Grant reference number | Author |
|---|---|---|
| National Institute on Drug Abuse | R01 DA038966 | Stephen Rayport |
| National Institute on Drug Abuse | R21 DA040443 | Susana Mingote Stephen Rayport |
| National Institute of Mental Health | T32 MH19970 | Abigail Kalmbach |
| National Institute of Mental Health | R01 MH113569 | Mark Ansorge |
| National Institute of Mental Health | R01 MH110556 | Rajeshwar Awatramani |
| National Institute of Neurological Disorders and Stroke | R01 NS101982 | Un Jung Kang |

The funders had no role in study design, data collection and interpretation, or the decision to submit the work for publication.

### Author contributions

Nao Chuhma, Conceptualization, Data curation, Formal analysis, Supervision, Investigation, Methodology, Writing—original draft, Project administration, Writing—review and editing; Susana Mingote, Data curation, Supervision, Investigation, Visualization, Writing—review and editing; Leora Yetnikoff, Samira Ztaou, Data curation, Investigation, Visualization, Writing—review and editing; Abigail Kalmbach, Data curation, Investigation, Methodology, Writing—review and editing; Thong Ma, Investigation, Methodology, Writing—review and editing; Anna-Claire Sienna, Sophia Tepler, Data curation, Investigation, Visualization; Jean-Francois Poulin, Resources, Visualization; Mark Ansorge, Rajeshwar Awatramani, Resources, Visualization, Writing—review and editing; Un Jung Kang, Resources, Investigation, Methodology, Writing—review and editing; Stephen Rayport, Conceptualization, Data curation, Supervision, Funding acquisition, Visualization, Project administration, Writing—review and editing

### Author ORCIDs

Nao Chuhma (iD) http://orcid.org/0000-0002-6846-3461
Susana Mingote (iD) http://orcid.org/0000-0002-0401-4317
Stephen Rayport (iD) http://orcid.org/0000-0001-9755-7486

### Ethics

Animal experimentation: Mice were handled in accordance with the guidelines of the National Institutes of Health Guide for the Care and Use of Laboratory Animals, under protocols approved by the Institutional Animal Care and Use Committee of New York State Psychiatric Institute, protocol NYSPI-1355.

Decision letter and Author response
Decision letter https://doi.org/10.7554/eLife.39786.026
Author response https://doi.org/10.7554/eLife.39786.027

## Additional files

### Supplementary files
• Transparent reporting form
DOI: https://doi.org/10.7554/eLife.39786.024

### Data availability
Source data are uploaded as supplements to figures.

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
