## [Decision Letter]

[**Editorial note:** This article has been through an editorial process in which the authors decide how to respond to the issues raised during peer review. The Reviewing Editor's assessment is that all the issues have been addressed.]

Thank you for submitting your article "Substantia nigra dopamine neurons evokes delayed excitation in lateral dorsal striatal cholinergic interneurons via glutamate cotransmission" for consideration by *eLife*. Your article has been reviewed by three peer reviewers, and the evaluation has been overseen by a Reviewing Editor and Gary Westbrook as the Senior Editor. The following individuals involved in review of your submission have agreed to reveal their identity: Antonello Bonci (Reviewer #1); Manuel Mameli (Reviewer #2). A third reviewer remains anonymous.

The Reviewing Editor has highlighted the concerns that require revision and/or responses, and we have included the separate reviews below for your consideration. If you have any questions, please do not hesitate to contact us.

Major concerns:

The statistical analysis is sophisticated and in most cases appropriate. The exception is the reporting of the electrophysiology data. The paper follows a long history of reporting this type of data on the basis of cells, rather than animals, as data points ("n"). Many reports are now demanding more transparency in the possible clustering (non-independence) of data points that come from individual animals. As a minimum, both number of cells and number of animals should be included with an indication of how statistics were calculated. Several recent references offer guidance in this regard: BMC Neurosci 16:94; https://doi.org/10.1371/journal.pone.0146721; J Neurosci 30:10601

Other suggestions from the reviews that the editor wishes to highlight:

1) The anatomical exploration in Figures 8-10 is a bit puzzling. There seems to be enrichment of VTA DA projections to ldStr over mdStr, but co-transmission turns out to be mediated by SN projections. Given that VTA projections to dorsal striatum appears to be a surprise, this would seem to warrant some follow up or additional explanation. What are these projections likely to be doing? Are they described in earlier literature?

2) Subsection “Substantia nigra DA neurons mediate glutamate cotransmission in the ldStr”, last paragraph, Figure 9. What about the double-labeled VTA neurons? Spread of virus is cited, but retrograde tracer shows VTA to ldStr projection, so the VTA projections need to be considered as possible sources of co-transmitting fibers.

3) Figure 9, subsection “Glutamate cotransmission from substantia nigra dopamine neurons”, last paragraph. What is the basis of the TH-/ChR2+ neurons in Figure 9? If leaky ChR2 expression, how can glutamate-only transmission be excluded as an explanation for the ldStr EPSCs? Were previously published validation procedures for INTRSECT employed?

4) The Discussion might benefit from further comment on the result of subregion selective slow excitation in ldStr, from a behavioral/systems perspective.

5) Figure 11. The schematic gives the somewhat misleading impression that DARs on SPNs are not activated by DA.

Separate reviews (please respond to each point):

*Reviewer #1:*

Chuhma et al. thoroughly characterize fast and slow effects of dopamine neurons stimulation in the lateral and medial portions of the dorsal striatum using pharmacology, cell tracing methods, and molecular techniques. This is one of those rare cases where a paper is as thorough and comprehensive from the beginning as it can be. For this reason, I have no major comments and I consider the paper suitable for publication in its current state.

*Reviewer #2:*

In this work the authors tested the hypothesis that dopamine neurons in substantia nigra present functional heterogeneity in controlling dorsal striatal neuronal populations, namely cholinergic interneurons. Combining genetic strategies with optogenetic based physiology the authors provide compelling evidence showing that dopamine innervation differently controls ChIn in medial an lateral portion of dorsal striatum. Specifically the lateral ChIn are modulated by glutamate co-release via mGluR signaling. This provides a delayed excitation of such neuronal population, which is absent in medially located neurons.

This is a very elegant experimental tour de force, showing that substantia nigra dopamine neurons activate different receptors located in distinct postsynaptic neurons. This opens new research lines to determine the behavioral relevance of such distinct dopamine modulation.

The data are solid, the paper very clear and nicely presented, and conclusions are in full supported by the data. I appreciate the precise controls designed for each main experiment. I believe this work can be published as it is and it would represent a very important step forward for the field.

I have two sole notes regarding this work. The first concerns the retrograde labeling experiments. The use of CTB makes the signal not striking, and injection sites are only relatively restricted. It would be outstanding if the authors could design a strategy allowing to identify whether or not DA neurons projecting to distinct striatal territory are part of the same population or not.

The second is the genetic analysis, which I find interesting per se, yet slightly detached from the rest of the paper and indeed weakly present also in the Discussion. The authors may want to consider to move this data-set either to supplementary material or even exclude it from the present manuscript.

Additional data files and statistical comments:

As stated already, although not a formal requirement, having an anatomical experiment allowing to understand whether the DA neurons projecting to medial and lateral dorsal striatum are part or not of the same neuronal population in the substantial migration would be beneficial.

*Reviewer #3:*

This paper represents a well-constructed, albeit somewhat incremental advance to our understanding of midbrain to striatal transmission. It helps to explain some apparently disparate observations in previous work on the topic of DA-glutamate co-transmission in dorsal striatum. Most of my suggestions for improvement can be considered minor. The weakest aspects of the paper are those characterizing the anatomical basis for the ldStr glutamate response.

The statistical analysis is sophisticated and in most cases appropriate. The one exception is in the reporting of the electrophysiology data. The paper follows a long history of reporting this type of data on the basis of cells, rather than animals, as data points. Many reports are now demanding more transparency in the possible clustering (non-independence) of data points that come from individual animals. Several recent references offer guidance on handling this: BMC Neurosci 16:94; https://doi.org/10.1371/journal.pone.0146721; J Neurosci 30:10601.

Figure 5 and subsection “Slow EPSCs in lateral dorsal striatum cholinergic interneurons”, first paragraph. Time controls are welcome and an example of thoughtful experimental design in this work. However, the requirement for 20 min incubation in the D1 antagonist is still puzzling. Is 20 min required if a higher concentration of SCH is used?

Subsection “Pharmacological analysis of slow EPSCs”, fifth paragraph. Is the SCH compound D1 selective or D1/D5 selective. Text says D1 antagonist, but figure says D1/5 blockade.

The anatomical exploration in Figures 8-10 is a bit puzzling. There seems to be enrichment of VTA DA projections to ldStr over mdStr, but co-transmission turns out to be mediated by SN projections. Given that VTA projections to dorsal striatum appears to be a surprise, this would seem to warrant some follow up or additional explanation. What are these projections likely to be doing? Are they described in earlier literature?

Subsection “Substantia nigra DA neurons mediate glutamate cotransmission in the ldStr”, last paragraph, Figure 9. What about the double-labeled VTA neurons? Spread of virus is cited, but retrograde tracer shows VTA to ldStr projection, so the VTA projections need to be considered as possible sources of co-transmitting fibers.

Figure 9 uses double conditional ChR2 expression, but optogenetic stimulation studies are not performed. Although there are caveats (below), this may help the arguments for co-transmission.

Figure 9, subsection “Glutamate cotransmission from substantia nigra dopamine neurons”, last paragraph. What is the basis of the TH-/ChR2+ neurons in Figure 9? If leaky ChR2 expression, how can glutamate-only transmission be excluded as an explanation for the ldStr EPSCs? Were previously published validation procedures for INTRSECT employed?

The CAV2 and INTRSECT strategies are fancy, but given that several caveats are noted for these, why not employ the older but better validated approach of combining retrograde labeling with immunostaining and/or ISH for vGLUT2 and TH?

Minor Comments:

Subsection “Synaptic currents underlying different ChI responses in the mdStr and ldStr”, first paragraph. Delete 'the' before 'sulpiride-sensitive'.

Subsection “DA neuron glutamate cotransmission in the lateral dStr”, first paragraph. Glutamate isolate not mentioned in the figure or legend. Please add for clarification.

Figure 2 legend. H-K. Please clarify that the color code from preceding panels applies here; done only in ldStr.

Figure 7 legend. Specify that recordings are done under GluR-isolating conditions.

Subsection “Slow EPSCs in lateral dorsal striatum cholinergic interneurons”, third paragraph. Check grammar.

The Discussion might benefit from speculation on the result of subregion selective slow excitation in ldStr, from a behavioral/systems perspective.

Figure 11. The schematic gives the somewhat misleading impression that DARs on SPNs are not activated by DA.

Additional data files and statistical comments:

See general comments. Number of animals should be dealt with in the electrophysiological data.

---

## [Author Response]

Reviewer #1:

Chuhma et al. thoroughly characterize fast and slow effects of dopamine neurons stimulation in the lateral and medial portions of the dorsal striatum using pharmacology, cell tracing methods, and molecular techniques. This is one of those rare cases where a paper is as thorough and comprehensive from the beginning as it can be. For this reason, I have no major comments and I consider the paper suitable for publication in its current state.

Thank you.

Reviewer #2:

In this work the authors tested the hypothesis that dopamine neurons in substantia nigra present functional heterogeneity in controlling dorsal striatal neuronal populations, namely cholinergic interneurons. Combining genetic strategies with optogenetic based physiology the authors provide compelling evidence showing that dopamine innervation differently controls ChIn in medial an lateral portion of dorsal striatum. Specifically the lateral ChIn are modulated by glutamate co-release via mGluR signaling. This provides a delayed excitation of such neuronal population, which is absent in medially located neurons.This is a very elegant experimental tour de force, showing that substantia nigra dopamine neurons activate different receptors located in distinct postsynaptic neurons. This opens new research lines to determine the behavioral relevance of such distinct dopamine modulation.The data are solid, the paper very clear and nicely presented, and conclusions are in full supported by the data. I appreciate the precise controls designed for each main experiment. I believe this work can be published as it is and it would represent a very important step forward for the field.I have two sole notes regarding this work. The first concerns the retrograde labeling experiments. The use of CTB makes the signal not striking, and injection sites are only relatively restricted.

In the CTB experiments, we sought to identify all DA neurons projecting to the medial or lateral dStr, and so we made larger injections, taking advantage of the spread of soluble CTB (unlike Fluoro-Gold or retrobeads). While we sought to fill the medial or lateral dStr, we were careful to avoid spread of CTB to the other side of the dStr. Figure 8A shows the differential targeting of the medial or lateral dStr. While spread to SPNs proximate to the injection sites made the injections appear larger, the limited SPN projections in the SNr showed that the injection sites were smaller than they appeared, confirming the differential targeting of the medial or lateral dStr.

It would be outstanding if the authors could design a strategy allowing to identify whether or not DA neurons projecting to distinct striatal territory are part of the same population or not.

Lerner et al., 2015, showed that different DA neuron populations project to the medial and lateral dStr, with CAV injections into the medial or lateral dStr and non-cell-selective cre injections into the SN. Our CAV2-FLEX-ZsGreen experiments show no projections to the medial dStr, consistent with this segregation. These results are also consistent with the recognized anatomy of SN DA neurons, whose axonal projections are restricted to a single striatal compartment, and do not branch to innervate other compartments (Matsuda et al., 2009, J Neurosci). So, medial and lateral dStr projecting DA neurons do form separate populations, except possibly for the small number projecting to the border between the medial and lateral dStr, which could be seen as projecting to both compartments.

The second is the genetic analysis, which I find interesting per se, yet slightly detached from the rest of the paper and indeed weakly present also in the Discussion. The authors may want to consider to move this data-set either to supplementary material or even exclude it from the present manuscript.

Presynaptic factors alone cannot explain why slow EPSCs are observed only in ChIs and not in SPNs in the lateral dStr, as both receive DA neuron glutamatergic inputs (Figure 3). Differences in effector expression in postsynaptic neurons could explain the difference. While, the gene expression analysis did not account for the mediolateral difference, it helps to account for cell-type specificity, and so deserves to be a main figure.

Additional data files and statistical comments:As stated already, although not a formal requirement, having an anatomical experiment allowing to understand whether the DA neurons projecting to medial and lateral dorsal striatum are part or not of the same neuronal population in the substantial migration would be beneficial.

We have now cited the Lerner et al. results, indicating that DA neurons projecting to the medial and lateral dStr are separate populations.

Reviewer #3:

This paper represents a well-constructed, albeit somewhat incremental advance to our understanding of midbrain to striatal transmission. It helps to explain some apparently disparate observations in previous work on the topic of DA-glutamate co-transmission in dorsal striatum. Most of my suggestions for improvement can be considered minor. The weakest aspects of the paper are those characterizing the anatomical basis for the ldStr glutamate response.The statistical analysis is sophisticated and in most cases appropriate. The one exception is in the reporting of the electrophysiology data. The paper follows a long history of reporting this type of data on the basis of cells, rather than animals, as data points. Many reports are now demanding more transparency in the possible clustering (non-independence) of data points that come from individual animals. Several recent references offer guidance on handling this: BMC Neurosci 16:94; https://doi.org/10.1371/journal.pone.0146721; J Neurosci 30:10601.

For the slice electrophysiology, we now report the number of animals, as well as the number of cells. For the FSCV, we now report the number of slices, in addition to the number of animals. We have added to the Statistical analysis section of the Materials and methods the limit on the number of cells recorded from any one animal.

Figure 5 and subsection “Slow EPSCs in lateral dorsal striatum cholinergic interneurons”, first paragraph. Time controls are welcome and an example of thoughtful experimental design in this work. However, the requirement for 20 min incubation in the D1 antagonist is still puzzling. Is 20 min required if a higher concentration of SCH is used?

The generally used concentration of SCH in brain slice studies is 10 µM SCH. Higher concentrations of SCH would shorten application times, but they also suppress calcium currents (Guenther, Wilsch and Zrenner, 1994).

Subsection “Pharmacological analysis of slow EPSCs”, fifth paragraph. Is the SCH compound D1 selective or D1/D5 selective. Text says D1 antagonist, but figure says D1/5 blockade.

Strictly speaking, SCH is a D1-like (D1/D5) receptor antagonist. Interestingly, ‘D1-like receptors’ is used, but ‘D1-like antagonist’ is rarely used; a D1-like receptor antagonist is referred to as a ‘D1 antagonist’. It is the same for D2 antagonists; sulpiride is a ‘D2 antagonist’, but is actually a ‘D2-like receptor antagonist’. We added the explanation that the D1 antagonist SCH blocks D1-like (D1/D5) receptors at the first mention of SCH (Figure 5).

The anatomical exploration in Figures 8-10 is a bit puzzling. There seems to be enrichment of VTA DA projections to ldStr over mdStr, but co-transmission turns out to be mediated by SN projections. Given that VTA projections to dorsal striatum appears to be a surprise, this would seem to warrant some follow up or additional explanation. What are these projections likely to be doing? Are they described in earlier literature?

Since the VTA-ldStr projections showed very little VGLUT2 expression, based on the CAV2-ZsGreen experiment (Figure 8), we presumed that the non-topographic projections were from non-glutamate cotransmitting DA neurons. However, those neurons may contribute to D2 or D5 responses in ChIs in the ldStr. The non-topographical projections are described in rat, as we cited, but have not been described in other species. We added this to the Discussion on “Glutamate cotransmission from substantia nigra dopamine neurons’ (first paragraph).

Subsection “Substantia nigra DA neurons mediate glutamate cotransmission in the ldStr”, last paragraph, Figure 9. What about the double-labeled VTA neurons? Spread of virus is cited, but retrograde tracer shows VTA to ldStr projection, so the VTA projections need to be considered as possible sources of co-transmitting fibers.

The CAV2-FLEX-ZsGreen experiment (Figure 8) showed that cotransmitting VTA DA neurons do not contribute significantly to the ldStr projections. Based on the CAV2 experiment results, we presume those glutamate cotransmitting VTA DA neurons follow the topographic projection pattern, and project to the nucleus accumbens. We added this to “Glutamate cotransmission from substantia nigra dopamine neurons” (second paragraph).

Figure 9 uses double conditional ChR2 expression, but optogenetic stimulation studies are not performed. Although there are caveats (below), this may help the arguments for co-transmission.

We utilized the INTRSECT as a further anatomical approach to visualize TH/VGLUT2 neurons. Regarding the demonstration of dopamine neuron glutamate cotransmission, the conditional VGLUT2 knockout (cKO) experiment is better controlled, as it depends on DAT (Figure 7), rather than TH, as the conditional driver.

Figure 9, subsection “Glutamate cotransmission from substantia nigra dopamine neurons”, last paragraph. What is the basis of the TH-/ChR2+ neurons in Figure 9? If leaky ChR2 expression, how can glutamate-only transmission be excluded as an explanation for the ldStr EPSCs? Were previously published validation procedures for INTRSECT employed?

The INTRSECT experiment and electrophysiological experiments used different transgenic mice with different drivers for DA-neuron specific expression of ChR2. For electrophysiology, we used DAT^IREScre^; floxSTOP-ChR2 (Ai32) double mutant mice. The DAT^IREScre^driver shows high specificity for DA neurons: about 98% of ChR2^+^ neurons immunostain for TH (Mingote et al., 2017). Therefore, a contribution from glutamate-only neurons is unlikely.

There are several possibilities for the lower specificity with the INTRSECT strategy. There may be more non-specific expression from the TH promoter than DAT (Lammel et al., 2015). The TH^2A-flpo^ mice used for the INTRSECT experiment show 93% specificity of reporter expression in DA neurons (Poulin et al., 2018), which is lower than the specificity with DAT^IREScre^. Expression specificity is presumably affected by the viral vectors as well. The AAV-hSyn-Con/Fon-hChR2-EYFP-WPRE virus we used may have a higher incidence of non-conditional recombination.

ChR2/TH colocalization is harder to visualize than staining with two cytoplasmic epitopes; since ChR2 is membrane-targeted, ChR2/TH double positive neurons are recognized as TH^+^ soma surrounded by ChR2^+^ membrane. Although we imaged both color channels in all z-stack planes carefully, some colocalized neurons may have been overlooked. Finally, TH expression varies in DA neurons (Bayer and Pickel, Brain Res, 1991; Stuber, Stamatakis and Kantak, 2015), so that weakly TH^+^ neurons may be missed. Lower specificity may be due to the combination of these factors.

Even with some non-specific ChR2 expression, examining TH^+^/VGLUT2^+^ fibers in the dStr, we see fibers in the ldStr, but only rare fibers in the mdStr. This supports the result of the CAV2 experiment, which was the major purpose of the INTRSECT experiment. We added this to the Discussion in “Glutamate cotransmission from substantia nigra dopamine neurons” (second paragraph).

The CAV2 and INTRSECT strategies are fancy, but given that several caveats are noted for these, why not employ the older but better validated approach of combining retrograde labeling with immunostaining and/or ISH for vGLUT2 and TH?

The purpose of the CAV2 and INTRSECT experiments was to reevaluate the results obtained by classical techniques. Although the new techniques have pitfalls; e.g. specificity of ChR2 expression with the INTRSECT strategy, as mentioned above, the well-established methods have their own limitations as well. Specifically, the issue was how to detect low levels of VGLUT2 expression in DA neurons. Although in situ hybridization (ISH) is a well-established and widely used technique, it does not have sufficient sensitivity to detect low-abundance messages. The CAV2 and INTRSECT enable better detection of VGLUT2 expression. Indeed, we found DA neurons in the SN, which were not previously detected by ISH; see also Poulin et al., 2018.

Minor Comments:Subsection “Synaptic currents underlying different ChI responses in the mdStr and ldStr”, first paragraph. Delete 'the' before 'sulpiride-sensitive'.

Done.

Subsection “DA neuron glutamate cotransmission in the lateral dStr”, first paragraph. Glutamate isolate not mentioned in the figure or legend. Please add for clarification.

Added to legend of Figure 7A.

Figure 2 legend. H-K. Please clarify that the color code from preceding panels applies here; done only in ldStr.

We added ‘ldStr’ labels to Figure 2H and 2J, and made traces in Figure 2J green to make the ‘color-coding’ clear.

Figure 7 legend. Specify that recordings are done under GluR-isolating conditions.

We added the description of the recording condition, as mentioned above.

Subsection “Slow EPSCs in lateral dorsal striatum cholinergic interneurons”, third paragraph. Check grammar.

Revised.

The Discussion might benefit from speculation on the result of subregion selective slow excitation in ldStr, from a behavioral/systems perspective.

We added further discussion about possible behavioral functions in “Functional Implications” (Discussion, last paragraph).

Figure 11. The schematic gives the somewhat misleading impression that DARs on SPNs are not activated by DA.

We added to the figure legend the statement that the schematic shows synaptic actions, and does not include modulatory actions of DA, through volume transmission. We do not mean to downplay volume transmission or modulatory actions of DA, but those are beyond the scope of the paper, which focuses on direct synaptic actions.

Additional data files and statistical comments:See general comments. Number of animals should be dealt with in the electrophysiological data.